# Characterizing the Discrete Geometry of ReLU Networks

**Blake B. Gaines**
Department of Computer Science
University of Connecticut
`blake.gaines@uconn.edu`

**Jinbo Bi**[*]
Department of Computer Science
University of Connecticut
`jinbo.bi@uconn.edu`

## Abstract

It is well established that ReLU networks define continuous piecewise-linear functions, and that their linear regions are polyhedra in the input space. These regions form a complex that fully partitions the input space. The way these regions fit together is fundamental to the behavior of the network, as nonlinearities occur only at the boundaries where these regions connect. However, relatively little is known about the geometry of these complexes beyond bounds on the total number of regions, and calculating the complex exactly is intractable for most networks. In this work, we prove new theoretical results about these complexes that hold for all fully-connected ReLU networks, specifically about their connectivity graphs in which nodes correspond to regions and edges exist between each pair of regions connected by a face. We find that the average degree of this graph is upper bounded by twice the input dimension regardless of the width and depth of the network, and that the diameter of this graph has an upper bound that does not depend on input dimension, despite the number of regions increasing exponentially with input dimension. We corroborate our findings through experiments with networks trained on both synthetic and real-world data, which provide additional insight into the geometry of ReLU networks. Code to reproduce our results can be found at `https://github.com/bl-ake/ICLR-2026`.

## 1 Introduction

Fully-connected networks with Rectified Linear Unit (ReLU) activations have become ubiquitous in recent years. These networks realize piecewise linear functions, with each "piece" defined on a polyhedron in the input space as illustrated in Fig. 1a (Grigsby & Lindsey, 2022; Grigsby et al., 2024). These functions can be incredibly complex and have universal approximation power if sufficiently wide or deep (Huang, 2020). Even work on one of the more basic questions about these networks—bounding the maximum number of regions defined by a given architecture—already spans over a decade (Montufar et al., 2014; Goujon et al., 2024). Since that work began, it has generated significant interest in other topics related to how ReLU networks divide the input space. The geometry of ReLU networks is defined both by the number of polyhedral regions and the way they connect to each other to form a polyhedral complex (Fig. 1b). Existing work that focuses on the number of regions does not describe their arrangement (Pascanu et al., 2014; Montúfar et al., 2022; Goujon et al., 2024). These works show that investigating specific properties of the complex (e.g., defining the boundaries of individual regions, calculating paths from one region to another) can be intractable because the number of regions grows exponentially with respect to the input dimension and network size (Hanin & Rolnick, 2019b). Here we attempt to fill the gap in the middle and establish general properties about the arrangement of these regions that hold regardless of network size and the actual values of network weights.

This work is motivated by the wide variety of research areas that leverage the polyhedral geometry of neural networks. These include explainability (Zhu et al., 2023; Hu, 2021), expressivity (Raghu et al., 2017), error prediction (Ji et al., 2022; Lehmann & Ebner, 2022; Daróczy, 2020),

---

[*]Corresponding Author

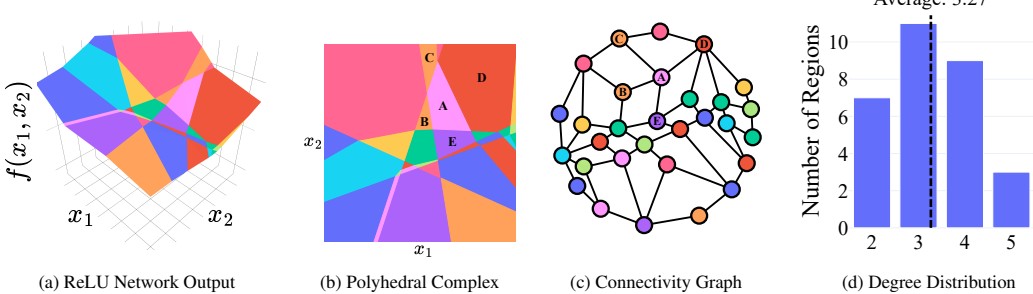

(a) ReLU Network Output  (b) Polyhedral Complex  (c) Connectivity Graph  (d) Degree Distribution

Figure 1: (a) An example ReLU network with a 2-dimensional input. (b) The corresponding polyhedral complex where region A has neighbors B, C, D, and E. (c) The connectivity graph where nodes represent regions and edges link neighboring regions (so region A has degree 4). (d) A histogram of the number of neighbors for each region, or equivalently the degrees of the connectivity graph.

robustness (Tran et al., 2019; Yang et al., 2020; Jamil et al., 2022), and even toxicity in large language models (Balestriero et al., 2024). It also allows the networks to be encoded as mixed-integer programs for verification (Botoeva et al., 2020; Cheng et al., 2017; Bunel et al., 2018) and inverse design (Ansari et al., 2022). The relationship between polyhedral regions and network activation states has also been applied to data compression (He et al., 2021), representation and clustering (Craighero et al., 2020), and open set detection (Jamil et al., 2023). A more detailed review of related work can be found in Appendix A.

Our analysis builds on the topological perspective of ReLU network geometry, and follows the same assumptions as Masden (2025). Our results are best expressed in terms of the complex's **connectivity graph** (Fig. 1c), where nodes correspond to polyhedral regions and edges exist between regions that have a shared face (Liu et al., 2023b). The degree of a node in the graph corresponds to the number of faces of its region, each of which connects to a unique neighboring region. Fig. 1d plots a histogram of the node degrees and shows the average degree of the connectivity graph in Fig. 1c. The diameter of this graph (the longest shortest-path distance between any pair of nodes in the graph) corresponds to the number of faces one has to cross to reach a polyhedron from any other polyhedron. Our work provides fundamental links between network architecture and connectivity graph topology. Recent work has analyzed the connectivity graph to characterize network properties such as VC-Dimension and the distribution of region volumes (Dhayalkar, 2025). Notably, the work in (Fan et al., 2024) includes several results bounding the average number of faces of the polyhedra from above, but with crucial assumptions for the ReLU networks (e.g., no bias terms or low rank in the first hidden layer's weight matrix) and their bounds are asymptotic with respect to the size of the network. In this work, we will bound the same quantity for networks regardless of architecture, both from above and (for networks with at least $d$ neurons in any configuration) below. We will then further characterize the complex by deriving similar bounds for the intersections of the regions, and bound the connectivity graph diameter from above and below, which provides additional insight into how the activation regions fit together. Our main contributions are as follows:

**Theoretical Properties**

A fully-connected ReLU network with input dimension $d$, maximum layer width $m$, and depth $\ell$ creates a polyhedral complex $\mathcal{C}$ in $\mathbb{R}^d$ that, with probability 1 (almost everywhere) over all possible network weights, satisfies:

1. The average degree of the connectivity graph is at most $2d$.

2. This average approaches the upper bound as the size of the network increases.

3. The diameter of the connectivity graph is bounded above by $(m+1)^\ell$ regardless of the value of $d$.

**Empirical Observations**

Experimental results with networks of different sizes trained on synthetic data and three benchmark datasets show that:

1. The average degree of the connectivity graph quickly approaches the upper bound as the size of the network increases.

2. The number of neighbors for every polyhedral region follows a unimodal distribution that skews right and peaks just below $2d$.

3. Regions that contain data points tend to be more connected on average compared to those that do not.

## 2 PRELIMINARIES

**Sign Sequences:** Let $f : \mathbb{R}^d \to \mathbb{R}^{\text{out}}$ be a fully-connected feedforward ReLU network in which each hidden neuron $i$ performs an affine transformation $w_i^T x^{p_i} + b_i$ on its input $x^{p_i}$ and then applies the ReLU activation, where $x^{p_i}$ is the output of the previous layer, and $w_i$ and $b_i$ are the trainable parameters of this neuron. Let $n$ denote the total number of hidden neurons in $f$. We start by introducing how, with probability 1 over possible network weights, the activation states of the neurons in a network can be uniquely represented by sign sequences (Masden, 2025). For a point $x$ in the input space, when passing through the network and reaching the $i$th neuron, if $w_i^T x^{p_i} + b_i > 0$, then the sign of this neuron $\mathcal{S}(x)_i = 1$ and the ReLU function outputs the value of $w_i^T x^{p_i} + b_i$; if the affine function is equal to or less than 0, then $\mathcal{S}(x)_i = 0$ or $-1$ respectively, and the ReLU outputs 0. Thus, $x$ receives a sign sequence $\mathcal{S}(x) \in \{-1, 0, 1\}^n$, a vector of length $n$ indicating the sign of each neuron's output before applying the activation function.

**Bent Hyperplanes:** If $i$ is in the first hidden layer, then $x^{p_i}$ is the network's input, and all inputs $x$ with $\mathcal{S}(x)_i = 0$ lie on the hyperplane $\{x \in \mathbb{R}^d : w_i^T x + b_i = 0\}$. When $i$ is a neuron from a later layer, the set of inputs with $w_i^T x^{p_i} + b_i = 0$ (i.e., $\mathcal{S}(x)_i = 0$) is more complex, because $x^{p_i}$ will have been computed by the continuous piecewise-linear function represented by the previous layers, and the input points for which $\mathcal{S}(x)_i = 0$ form a level set of this function. We call this set the neuron's Bent Hyperplane (BH) following convention (Hanin & Rolnick, 2019b; Masden, 2025), although unlike a hyperplane, a BH can intersect itself and even be disconnected.

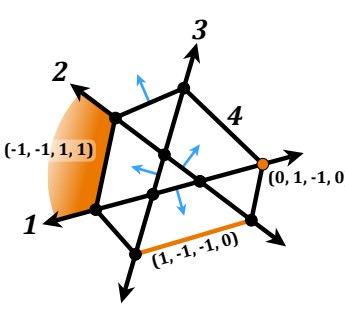

Figure 2: A polyhedral complex.

Fig. 2 shows an example of BHs created by a network with an input of $d = 2$ and 2 hidden ReLU layers where 3 neurons are in layer 1 (corresponding to hyperplanes, which we also call BHs 1–3 for notational convenience) and 1 neuron in layer 2 (corresponding to BH 4). Each blue arrow in Fig. 2 indicates the orientation of $\text{BH}_i$, pointing towards the area where the output of the neuron $i$ is positive (i.e., $\mathcal{S}(x)_i = 1$). BHs 1–3 intersect to form 7 regions. BH 4 intersects 6 of them, and in each area, BH 4 is a segment of a hyperplane. Whenever BH 4 crosses a BH $j$ in the first layer, one of the entries of its input $x^{p_i}$ changes its activation state (switching between $w_j^T x + b_j$ and 0). This causes the BH to bend. In this example, it eventually intersects itself.

The BH of neuron $i$ (where $\mathcal{S}(x)_i = 0$) forms a boundary that divides the input space into two parts where $\mathcal{S}(x)_i = 1$ and $\mathcal{S}(x)_i = -1$. Thus, the sign sequence of a region can also be interpreted as a list describing which "side" of each BH it lies on. All of the BHs of neurons in previous layers partition the input space into disjoint regions, in each of which, $w_i^T x^{p_i} + b_i$ collapses into an affine function in the input space (say $\Phi_i^T x + \beta_i$ for some $\Phi_i \in \mathbb{R}^d$ and $\beta_i \in \mathbb{R}$), so the segment of the BH within an area lies on a hyperplane in the input space $\Phi_i^T x + \beta_i = 0$. This hyperplane subdivides the regions it intersects into two smaller regions that differ in the activation state of neuron $i$. For instance, Fig. 2 shows that BH 4 divides the 6 regions formed by intersections of BHs 1–3 into 12 smaller regions.

From this point forward, all statements about ReLU networks will make the same assumptions as in (Masden, 2025) to avoid degenerate weight assignments. These will ensure that at most $d$ BHs intersect at a point, and that the sections of BHs are never perfectly parallel to each other. As proven in that work, these assumptions will hold on all but a measure-zero set of parameter assignments for a given architecture. Appendix B provides rigorous definitions of these assumptions.

**Polyhedral Complex:** A polyhedral complex is a set of polyhedra (cells) that is closed under intersection (e.g., the complex in Fig. 2 includes not only the polygons as elements but also the line segments where pairs of polyhedra intersect and the vertices where four polygons intersect), and taking faces (e.g., the complex in Fig. 2 includes the line segments enclosing each polyhedron and the vertices enclosing each line segment). A $k$-cell is an element of a polyhedral complex with affine span $k \leq d$, that is, a polyhedral set whose elements span a $k$-dimensional affine subspace of $\mathbb{R}^d$. BHs of a neural network form the boundaries of the network's $d$-cells (the maximal regions in which the network's mapping is affine), which intersect to generate the polyhedral complex $\mathcal{C}$ of the network (Grigsby & Lindsey, 2022). Within each cell of $\mathcal{C}$, the network's behavior is affine. When

$k < d$, the $k$-cells of this complex are each contained by the intersection of $d - k$ BHs. For the complex in Fig. 2, the orange 2-cell is not contained in any BHs because $d - k = 0$, a 1-cell is contained by a BH (e.g., the orange line segment is part of BH 4) and a 0-cell is contained in the intersection of two BHs (e.g., the highlighted vertex is the intersection of BHs 1 and 4). Accordingly, the BH of neuron $i$ can be considered as the union of the $(d - 1)$-cells with a single 0 in the $i$th position of their sign sequences. For example, in Fig. 2, BH 4 is formed by the ring of six line segments (1-cells) with a 0 in position 4 of their sign sequences. We define the "faces" of a $k$-cell as the $(k-1)$-cells it contains (these are often called "facets" in the relevant literature). In the connectivity graph, the $k$-cells of $\mathcal{C}$ are represented by $(d - k)$-hypercube subgraphs, and the collection of edges corresponding to the 1-cells of each BH form a cut.

An important property of a network's canonical polyhedral complex is that if it is restricted to the cells lying in or on one side of a single neuron's BH (or equivalently, an element of the sign sequences is fixed), the resulting substructure is still a polyhedral complex. Although it no longer corresponds to a ReLU network, we still call it a ReLU complex because it is still a polyhedral complex with cells defined by BHs, so several of the results in the next section will still apply. In the following discussion, the dimension of a ReLU (sub)complex will refer to the maximum dimension of its cells instead of the dimension of the ambient space in which it is embedded (e.g. a BH of a 2-dimensional ReLU complex is a 1-dimensional ReLU complex).

**Sign Sequence Complex:** The polyhedral complex can be described in terms of sign sequences, because in the interior of any $k$-cell, the sign sequence of every point is exactly the same. Furthermore, the sign sequences of a $k$-cell contain exactly $d - k$ zero elements corresponding to the $d - k$ BHs whose intersection contains the cell as a subset and $k$ nonzero elements corresponding to BHs that either contain a single face of the cell or do not intersect the cell at all. For example, in Fig. 2, the orange 1-cell with sign sequence $(1, -1, -1, 0)$ is contained by BH 4 so it has 0 in this position, its faces are the vertices on each end contained by BHs 2 and 3 respectively, and BH 1 does not intersect the cell at all, so all three are nonzero in the sign sequence. With the aforementioned basic assumptions on the network weights, the work in (Masden, 2025) proves that every cell of a ReLU complex has a unique sign sequence, that is, the mapping from cells to sign sequences $\mathcal{S} \colon \mathcal{C} \to \{-1, 0, 1\}^n$ is well-defined and injective. The connectivity graph can be equivalently defined in terms of the sign sequence complex, with nodes for the sign sequences of $d$-cells (the ones with no zeros) and edges between sequences that differ by one element.

## 3 THEORETICAL RESULTS ON NETWORK GEOMETRY

We examine how the cells in a ReLU complex are connected with each other, how many neighbors a cell can have on average, and whether or not the number of neighbors varies with respect to the depth and width of the network. Proof outlines are given here while detailed proofs are in Appendix B. The cells in a ReLU complex and the number of their faces may depend on the specific network architecture and values of the network weights. However, we prove that the average number of neighbors for any polyhedral region can be upper bounded by $2d$ regardless of the depth and width of the network. In a ReLU complex, counting the faces of cells is the same as counting their neighbors. For a $d$-cell, each face is contained in a unique BH, and across each face is a unique neighboring polyhedron that has the same sign sequence except that the sign corresponding to the crossed BH is flipped. More generally, we prove the following theorem for any $k$-cells of a ReLU complex.

**Theorem 3.1.** *For a ReLU complex in $d$-dimensional input space, the average number of faces of a $k$-cell is at most $2k$ for $k = 1, 2, \ldots, d$.*

An earlier work proves this theorem for hyperplane arrangements (Fukuda et al., 1991), which only applies to the polyhedral complexes of single-layer networks, but the proof does not generalize to deep ReLU network complexes formed by BH arrangements. We employ the 1-1 mapping between cells and sign sequences to prove that the theorem also holds true for complexes of deep ReLU networks.

Let $\mathcal{C}$ be a ReLU complex corresponding to a network $f$. Denote the BH of a neuron $i$ by $h_i$, which contains a set of $k$-cells of $k < d$, specifically $h_i = \{c \in \mathcal{C} \colon \mathcal{S}(c)_i = 0\}$. Then, we use $\mathcal{C} - h_i$ to denote the complex that results from removing all cells contained in the BH of neuron $i$ and joining all pairs of cells sharing one of the faces that were removed. Fig. 3a illustrates $\mathcal{C} - h_i$ with $\mathcal{C}$ from Fig. 2 and BH 4 as $h_i$. The connectivity graph of $\mathcal{C} - h_i$ can be obtained by contracting every edge

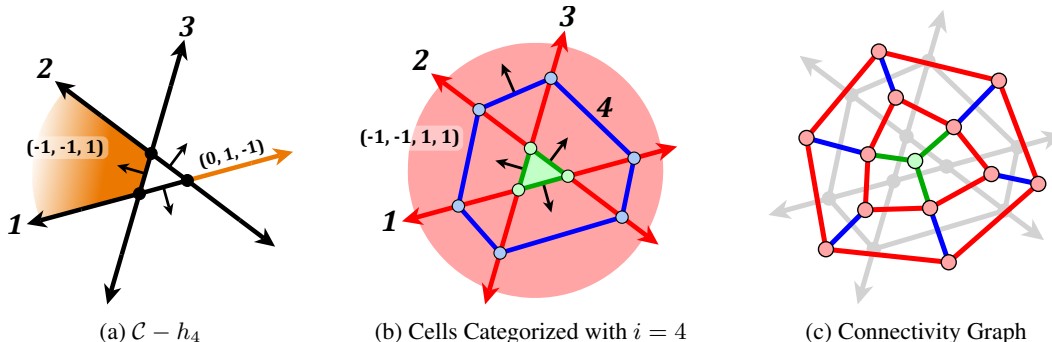

(a) $\mathcal{C} - h_4$  (b) Cells Categorized with $i = 4$  (c) Connectivity Graph

Figure 3: (a) The complex in Fig. 2 with BH 4 removed. (b) BH 4 is added and cells are categorized according to Lemma 3.2 with those in Categories 1, 2, and 3 shown in blue, green, and red, respectively. (c) Connectivity graph with nodes and edges colored according to their corresponding cells.

corresponding to a $(d - 1)$-cell contained in $h_i$ (i.e., removing each edge and combining the two end nodes into one, keeping their connections to other nodes in the graph). As a result, every two $k$-cells previously split by $h_i$ are now fused into a single new $k$-cell. Any cells that do not intersect $h_i$ remain the same in the new complex $\mathcal{C} - h_i$.

**Lemma 3.2.** *For any neuron $i$ in $f$, each $k$-cell $c$ of $\mathcal{C}$ falls into exactly one of the following categories:*

*Category 1: $c$ is a cell of $h_i$*
*Category 2: $c$ is a cell of $\mathcal{C} - h_i$*
*Category 3: $c$ is one of the two $k$-cells formed when a $k$-cell in $\mathcal{C} - h_i$ is separated by $h_i$*

We outline our full proof here. In the sign sequence of $c$, the element $\mathcal{S}(c)_i$ is either zero or nonzero. If it is zero, then $c$ can only be in Category 1. If it is nonzero, we need to check if $c$ intersects $h_i$. This is the case if changing the cell's sign sequence so that $\mathcal{S}(c)_i = 0$ yields a new sign sequence that matches a Category 1 cell in $\mathcal{C}$. Then, flipping $\mathcal{S}(c)_i$ yields the sign sequence of a $k$-cell neighbor of $c$, so $c$ is in Category 3. Otherwise, if the new sign sequence is not an element of $\mathcal{C}$, then $h_i$ does not contain $c$ or form a face of $c$, so $c$ is in Category 2.

Fig. 3b and Fig. 3c show the categorization of $k$-cells ($k = 0, 1, 2$) in the polyhedral complex from Fig. 2 when $h_i$ corresponds to BH 4. The blue line segments and vertices in Fig. 3b are the Category 1 1- and 0-cells respectively, and the 4th element of their sign sequences is 0. The green 2-, 1-, and 0-cells are in $\mathcal{C} - h_4$ because they do not change when $h_i$ is removed. For such a cell, if we zero out the 4th element of their sign sequence (e.g., the 2-cell in the center with a sequence $(-1, -1, -1, -1)$), the resulting sequence (e.g., $(-1, -1, -1, 0)$) does not exist in $\mathcal{S}(\mathcal{C})$. The red 2- and 1-cells are in Category 3 because $h_4$ contains one face of each of those cells. As an example, the leftmost 2-cell has the sign sequence $(-1, -1, 1, 1)$, and after setting $\mathcal{S}(c)_4 = 0$, $(-1, -1, 1, 0)$ corresponds to the line segment of $h_i$ forming its right boundary. Note that the proof of Lemma 3.2 also works for ReLU subcomplexes formed by fixing one of the sign sequence elements, and these subcomplexes cannot include only half of a pair of Category 3 cells since their sign sequences can only differ at the position of the removed BH $i$. Alternate colorings for the same complex based on different choices of $i$ and restrictions to different subcomplexes are included in Appendix C.

If $h_i$ is a neuron from the last ReLU layer, the complex $\mathcal{C} - h_i$ corresponds to another ReLU network, but directly removing a neuron from an early layer may not result in a complex corresponding to any ReLU network. As a result, we can break down the problem of counting cells (including faces) in the ReLU complex by iteratively removing neurons starting from the last layer and counting the number of cells that disappear. Let $N_k(\mathcal{C})$ be the total number of $k$-cells in $\mathcal{C}$. Based on Lemma 3.2, $N_k(\mathcal{C})$ equals the sum of the numbers of $k$-cells in each category. To evaluate $N_k(\mathcal{C})$, we first count the $k$-cells in $h_i$ and $\mathcal{C} - h_i$ separately, and then double count those in $\mathcal{C} - h_i$ that are split by $h_i$. To count the split cells, we can just count $(k - 1)$-cells in $h_i$, which each divide one of them. For example, compare Fig. 3a and Fig. 3b. The six 1-cells in $h_4$ split the six 2-cells in Fig. 3a to add six more 2-cells in Fig. 3b. Similarly, the six 0-cells in $h_i$ also split six 1-cells.

**Lemma 3.3.** *For $k = 1, 2, \ldots, d$,*

$$N_k(\mathcal{C}) = N_k(h_i) + N_k(\mathcal{C} - h_i) + N_{k-1}(h_i). \tag{1}$$

The first term in the sum accounts for the Category 1 $k$-cells in $\mathcal{C}$, the second term accounts for all the Category 2 cells and half the Category 3 cells, and the third term accounts for the other half of the Category 3 cells. There are no Category 1 $d$-cells in $\mathcal{C}$ because $h_i$ by definition only contains cells up to dimension $d-1$, so when $k=d$, the first term $N_k(h_i)$ is always 0. The two lemmas lead to the following special case of Theorem 3.1 for $k=d$:

**Theorem 3.4.** *[Upper Bound] The average number of faces of a $d$-cell of $\mathcal{C}$ in $\mathbb{R}^d$ (i.e., the average degree of the connectivity graph) is at most $2d$.*

Here we provide an outline of our proof (see Appendix B for detailed proof). Each $(d-1)$-cell forms a face between two $d$-cells in $\mathcal{C}$ because the single 0 in its corresponding sign sequence can be set to $-1$ or $1$ to get the sign sequences of the two $d$-cells. Thus, the sum of the numbers of faces of all $d$-cells is just twice the total count of $(d-1)$-cells in $\mathcal{C}$, so the average number of faces of a $d$-cell is $\frac{2N_{d-1}(\mathcal{C})}{N_d(\mathcal{C})}$. Using Lemma 3.3, we prove $\frac{2N_{d-1}(\mathcal{C})}{N_d(\mathcal{C})} \leq 2d$ by mathematical induction on the number of BHs $n$ in the complex and $d$. By assuming that the upper bound holds for $(n-1, d-1)$ and $(n-1, d)$, we prove that the upper bound holds for any complex with $(n, d)$. The proof of Theorem 3.1 then follows by applying the lemmas to groups $k$-cells whose sign sequences have exactly $d-k$ zeros at the same positions, that is, restricting the complex to the intersections of $d-k$ BHs and applying the lemmas to the resulting subcomplex.

It is more straightforward to establish the following lower bound on the degree of individual nodes, which then bounds the overall average degree of the ReLU complex graph.

**Theorem 3.5.** *[Lower Bound] If a ReLU network has $n_1$ neurons in the first hidden layer, every $d$-cell of $\mathcal{C}$ has at least $\min(n_1, d)$ neighbors, and thus the average degree of the connectivity graph is at least $\min(n_1, d)$.*

### 3.1 Asymptotic Behavior

To study how connectivity properties change as network size increases, we can create sequences of networks by adding new ReLU neurons to the last layer or a new layer after it. We use $\mathcal{C}_n$ to denote the complex after $n$ neurons have been added. We characterize these sequences with the following theorems, which show that the average number of faces grows monotonically and that the bound in Theorem 3.1 is tight respectively.

**Theorem 3.6.** *The average number of faces of $d$-cells in $\mathcal{C}_n$ increases monotonically in terms of $n$.*

**Theorem 3.7.** *Let $f$ be a shallow network that has only one hidden layer with $n$ nodes. When $n$ goes to infinity, the average number of faces of its $d$-cells converges exactly to $2d$. That is,*

$$\lim_{n \to \infty} \frac{2N_{d-1}(\mathcal{C}_n)}{N_d(\mathcal{C}_n)} = 2d.$$

In our experiments in Section 5, we observe that the average number of faces also appears to approach $2d$ as the depth of the network increases.

### 3.2 Bounds on Connectivity Graph Diameter

Let $\ell$ be the total number of hidden layers (depth), $m_j$ be the number of nodes in layer $j$, $j = 1, \ldots, \ell$, and $m = \max\{m_1, \cdots, m_\ell\}$ (width). We find that,

**Theorem 3.8.** *The diameter $D$ of the connectivity graph is $\Omega\left(\frac{\ln(N_d(\mathcal{C}))}{\ln(n)}\right)$ and $O\left(m^\ell\right)$.*

The lower bound (in $\Omega$) agrees with the intuition that diameter increases with the number of regions in the complex. Although it appears as though increasing the number of neurons in the network might reduce diameter by increasing $\ln(n)$, actually $\ln(N_d(\mathcal{C}))$ grows much faster with $n$ regardless of architecture, so the $\ln(n)$ term is just attenuating the growth of this lower bound. The upper bound (in $O$) may rarely be reached in practice, but it is interesting in that it does not have to depend on the input dimension $d$, even though the number of the network's regions increases exponentially with $d$. We also find that this is empirically true in Section 5, as when we fix network architecture and only change the input dimension, the diameter of the resulting complexes grows almost identically.

## 4  ALGORITHM FOR CALCULATING POLYHEDRON BOUNDARIES

To define the complex, it will be necessary to map sign sequences to their polyhedra defined by intersections of half-spaces, i.e., systems of linear inequalities of the form $\Phi_i^T x + \beta_i \leq 0$ for $i \in f$. We are only concerned with the sign sequences of $d$-cells, which do not contain any zeros, so that each neuron provides exactly one inequality to our system defining the polyhedron. Let $s$ be such a sign sequence and the column vector $s^{(j)}$ be the portion of $s$ that contains only signs for the neurons in layer $j$. Let $W^{(j)} \in \mathbb{R}^{j_{\text{out}} \times j_{\text{in}}}$ and $b^{(j)} \in \mathbb{R}^{j_{\text{out}}}$ denote the weights and biases in layer $j$ of the network with input dimension $j_{\text{in}}$ and output dimension $j_{\text{out}}$. We can define our polyhedron by using the following formulas to calculate the inequalities layer by layer.

Half-spaces for current layer          Picking half-space signs          Mask for inactive neurons in the previous layer

$$\Phi^{(j)} = \text{diag}\left(s^{(j)}\right) \ W^{(j)} \ \text{diag}\left(\text{ReLU}\left(s^{(j-1)}\right)\right) \ \Phi^{(j-1)} \tag{2}$$

Current layer's weights ↑          Half-spaces for the previous layer ↑

$$\beta^{(j)} = \text{diag}\left(s^{(j)}\right)\left(W^{(j)}\text{diag}\left(\text{ReLU}\left(s^{(j-1)}\right)\right)\beta^{(j-1)} + b^{(j)}\right) \tag{3}$$

At the initial stage, $\Phi^{(0)} = I_d$, $b^{(0)} = 0_{(d \times 1)}$, and $s^{(0)} = \mathbf{1}_{d \times 1}$. The first term on the right-hand side ensures that the inequality of each half-space is always in the same direction, regardless of whether the neuron is active or inactive. This allows us to concatenate $\Phi^{(j)}$ and $\beta^{(j)}$ from each layer to get the full linear system, $\Phi x + \beta \leq 0$.

### 4.1  ENUMERATING POLYHEDRA

To enumerate the maximal polyhedra and obtain their connectivity graph $G = (V, E)$, we will employ breadth-first search (BFS). We describe our exact method in Algorithm 1. Starting with a valid sign sequence $s$, which can be found by passing any point through the network, we enumerate its neighbors and add them to our graph. Neighbors are polyhedra that can be reached by crossing a single BH, so their sign sequences have the opposite sign from the original polyhedron in exactly one position $i$, denoted as $s_{-i}$. To find the neighbors, we can calculate $\Phi_s$ and $\beta_s$ as described in the previous section and determine which inequalities actu-

---

**Algorithm 1** Construction of the Connectivity Graph

1: **Input:** Trained network $f$, sign sequence $s$
2: $Q, V, E \leftarrow \{s\}, \{s\}, \emptyset$
3: **while** $Q$ is not empty **do**
4:    $s \leftarrow \text{pop}(Q)$
5:    **for** $i \in \{0, \dots, n\}$ **do**
6:       **if** $\text{SOLVELP}(-\Phi_{s_i}, \Phi_s, \beta_s + e_i) \geq \beta_{s_i}$ **then**
7:          add (s, $s_{-i}$) to $E$
8:          **if** $s_{-i} \notin V$ **then** add $s_{-i}$ to $Q$ and $V$ **end if**
9:       **end if**
10:    **end for**
11: **end while**
12: **return** $(V, E)$

---

ally form the boundary of the polyhedron. We check the redundancy of each inequality by solving an LP, performed by the SOLVELP subroutine on line 6, which is given the arguments $\Phi_s$ for the constraint coefficient matrix, $\beta_s + e_i$ ($\beta_s$ with 1 added to the $i$th element to relax this constraint) for the constraint offset, and $-\Phi_{s_i}$ for the objective function coefficients to maximize in the direction of the relaxed constraint. This LP is explained further in Appendix D. Non-redundant constraints will be violated by the optimal solution to this LP, i.e., $\Phi_{s_i}^T x > \beta_{s_i}$, meaning that $s_{-i}$ gives the sign sequence of the neighbor of $s$ across the BH of neuron $i$. For each neighbor, we add the edge between $s$ and $s_{-i}$ to the graph (line 7), and if we have not reached $s_{-i}$ before, we add it to both the graph and the search queue (line 8). In the next iteration, we can pop a new sign sequence from the queue and repeat the same process.

This traversal of polyhedra is similar to several previous works (Xu et al., 2022; Liu et al., 2023a;b), specifically the BFS in (Xu et al., 2022), but we take the additional step of building up the connectivity graph of the polyhedral complex over the course of the search by recording when faces are shared with already-found polyhedra. Furthermore, when determining whether or not an element of a sign sequence can be flipped to produce another valid sign sequence, we follow (Zhang & Wu, 2019; Fukuda, 2004) and slightly relax the corresponding inequality to reduce errors arising from insufficient numerical precision.

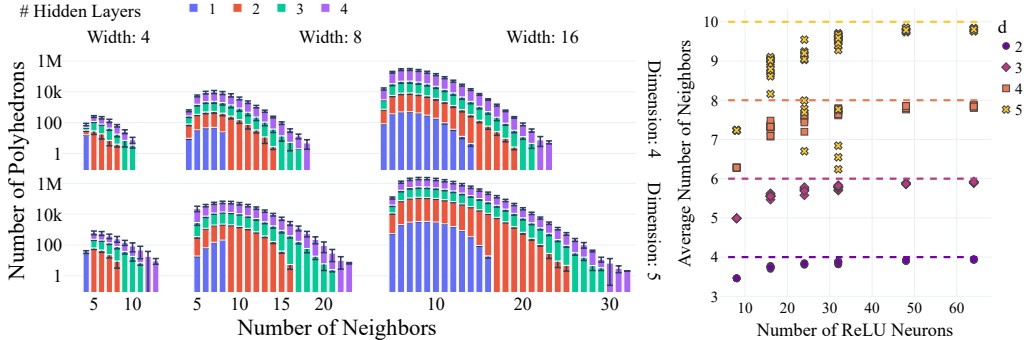

Figure 4: (Left) Distributions for the number of faces of polyhedra in ReLU networks trained on synthetic data. Each colored bar shows the number of polyhedra with a given number of faces in complexes of networks with a specific width, depth, and input dimension, averaged across 5 initializations of network weights for 5 different training datasets with standard deviation shown by the black bars. (Right) The mean of each distribution versus the number of neurons in the network, colored by dimension. Dotted lines represent upper bounds for the networks with different $d$.

## 5 EXPERIMENTS

To further understand the structure of ReLU complexes, we use Algorithm 1 to enumerate polyhedra for a number of neural networks and construct their connectivity graphs. We then discuss how data tends to be distributed across the complex of networks after training. Additional details about the experimental setup and trained networks can be found in Appendix F. Code, data, and models used in all experiments can be found at `https://github.com/bl-ake/ICLR-2026`.

### 5.1 UNDERSTANDING THE BOUNDS

We start by training a number of networks on clustering problems generated from three isotropic Gaussians with unit variance and centers selected uniformly at random within a hypercube around the origin with side length 10. We vary input dimension $d$, the number of hidden layers, and the width of each hidden layer. For each combination of hyperparameters, we perform five experiments with newly generated datasets, and perform an exhaustive search to compute the full complex of the network and obtain information about every region. Summary statistics for the polyhedra can be found in Table 1, and the distributions of neighbor counts are visualized in Fig. 4. The average number of neighbors in every complex is below the upper bound of $2d$, with the overall distribution being unimodal and skewed right. We also plot the estimated connectivity graph diameter versus the upper bound from Section 3.2 in Fig. 5. We estimate the actual diameter of the complexes

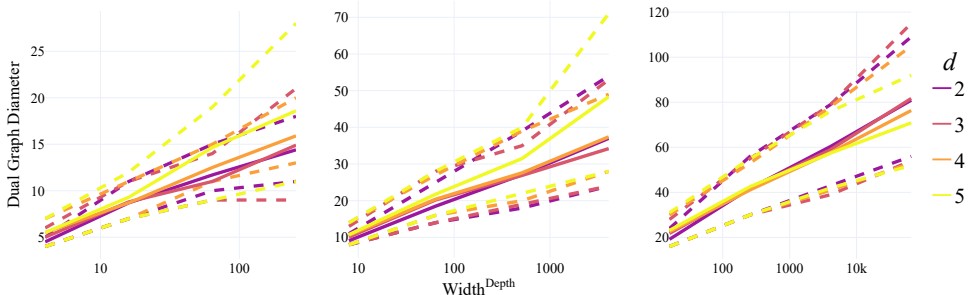

Figure 5: Connectivity graph diameter vs theoretical upper bound. At each distinct value of the theoretical upper bound (abscissa), the actual diameter of 5 network complexes was estimated as described in Section 5.1. Each pair of dotted lines also encloses all (non-estimated) values of the connectivity graph diameters from every experiment. Each subfigure shows networks with fixed width $m$ and depth $1 \le \ell \le 4$. The widths are 4 (left), 8 (middle), and 16 (right).

Table 1: Summary statistics for the distributions in Fig. 4 with four dimensions (left) and five dimensions (right). Diameter for each complex is estimated as described in Section 5.1. Non-degenerate depth-1 networks always have the same number of polyhedra because their BHs are all just hyperplanes (Buck, 1943).

| Depth | Width | # Polyhedrons | Average Degree | Diameter | # Polyhedrons | Average Degree | Diameter |
|---|---|---|---|---|---|---|---|
| 1 | 4 | 16.00±0.00 | 4.00±0.00 | 5.50±0.00 | 16.00±0.00 | 4.00±0.00 | 5.50±0.00 |
|  | 8 | 163.00±0.00 | 6.28±0.00 | 10.60±0.42 | 219.00±0.00 | 7.23±0.00 | 10.75±0.26 |
|  | 16 | 2517.00±0.00 | 7.32±0.00 | 22.50±0.35 | 6884.87±0.35 | 9.02±0.00 | 23.17±0.41 |
| 2 | 4 | 72.60±22.70 | 5.21±0.31 | 8.70±0.76 | 89.50±19.78 | 5.34±0.25 | 9.30±0.63 |
|  | 8 | 2244.80±630.08 | 7.25±0.18 | 20.40±0.74 | 5802.60±1146.10 | 8.77±0.27 | 21.75±1.06 |
|  | 16 | 42243.00±8608.12 | 7.72±0.06 | 41.10±0.65 | $2.69\times10^5$±48746.09 | 9.61±0.05 | 42.57±1.53 |
| 3 | 4 | 227.60±42.21 | 5.85±0.16 | 12.50±0.61 | 389.60±188.02 | 5.65±0.41 | 14.65±2.32 |
|  | 8 | 9340.80±3325.81 | 7.47±0.17 | 27.60±2.25 | 36591.20±12085.54 | 8.54±0.93 | 31.50±2.33 |
|  | 16 | $2.23\times10^5$±72142.81 | 7.82±0.04 | 57.70±2.46 | $1.78\times10^6$±$1.94\times10^5$ | 9.78±0.04 | 57.44±1.47 |
| 4 | 4 | 448.00±119.14 | 6.17±0.12 | 15.90±1.19 | 1206.70±1154.00 | 5.50±0.78 | 18.60±3.98 |
|  | 8 | 35767.20±9493.85 | 7.70±0.06 | 37.40±1.29 | $1.82\times10^5$±79768.45 | 8.25±1.38 | 48.35±12.25 |
|  | 16 | $6.24\times10^5$±96311.16 | 7.85±0.03 | 76.35±4.56 | $5.03\times10^6$±$1.07\times10^6$ | 9.80±0.03 | 70.88±1.19 |

by bounding each one above and below using the corresponding algorithms from (Magnien et al., 2009) and taking the midpoint. To be clear, the asymptotic bounds derived in Section 3.2 were not used to make this estimate. Across all experiments, the diameter estimates for networks with the same depth and width were almost identical across different input dimensions. Although the upper bound is rarely reached, the logic that it should be independent of input dimension appears to hold in practice. Furthermore, when width is fixed, the diameter appears to grow logarithmically with respect to our theoretical upper bound. Additional summary metrics for the complexes and results for $d \in \{2, 3\}$ can be found in Appendix G.

## 5.2 TRAINING DATA AND POLYHEDRA

We observe a difference in the distributions of neighbor counts between polyhedra that contain data and those that do not. We test networks trained on three datasets: California Housing (CC 0 License) (Kelley Pace & Barry, 1997), MNIST (CC BY-SA 3.0 License) (Deng, 2012), and CIFAR10 (MIT License) (Krizhevsky, 2009), and achieve reasonable performance for each (AUC above 0.9 or $R^2$ above 0.6). We examine the last 3 layers of 8 neurons for MNIST and 2 layers of 64 neurons for CIFAR10 on a lower-dimensional hidden representation rather than the input, 5 dimensions for MNIST and 10 for CIFAR10. For California Housing, we calculate the complex of the entire network. Details about the datasets and networks can be found in Appendix F. Algorithm 1 was used to identify all polyhedra in the complex for MNIST. For the California Housing and CIFAR10 datasets, complete enumeration of the network complex was intractable, so the search was terminated after traversing 8 million polyhedra. We then randomly sample 10,000 points from the training data. If a data point does not lie in one of the polyhedra found in the initial search, we calculate the new polyhedron that contains the point and add it to the 8 million that were already found. The distributions of neighbor counts for these complexes can be found in Fig. 6. Across all datasets, the neighbor counts for polyhedra containing training data tend to be higher than the upper bound for the average neighbor count of all polyhedra. Since the number of faces of any polyhedron is bounded above by $n$, this necessarily reduces the rightward skew of the distribution as well.

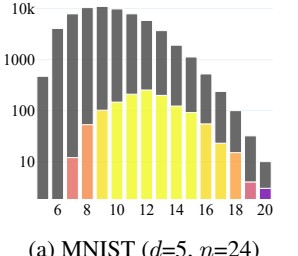

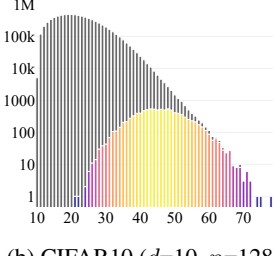

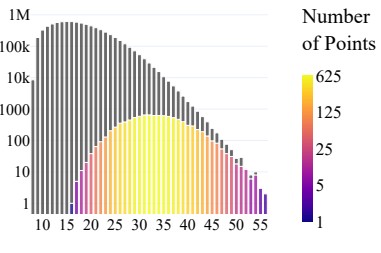

(a) MNIST ($d$=5, $n$=24)    (b) CIFAR10 ($d$=10, $n$=128)    (c) CA Housing ($d$=8, $n$=128)

Figure 6: Histograms of polyhedron neighbor counts (i.e., the number of polyhedra that have a specific number of neighbors) for polyhedra that do not contain training data (gray) and ones that do (colored by total number of data points contained in those polyhedra).

We also examine how neighbor counts vary according to whether polyhedra are bounded or unbounded, with results shown in Fig. 7. In all three experiments, we observe that polyhedra with higher numbers of neighbors are more likely to be unbounded (darker colors toward the right of each histogram, with the exception of polyhedra with $d$ neighbors shown by the leftmost bars, which are always unbounded). In addition, we find that the proportion of unbounded polyhedra in data-containing regions is higher than the overall proportion for both classification tasks (the top two histograms show darker bars than the corresponding bottom figures) but lower for the regression task (the top histogram bars have lighter colors than the bottom). For the classification tasks, the network may have to focus its complexity on the spaces between classes of data points where it has to draw the decision boundary, leaving more of the data points themselves on the outer (unbounded) regions of the complex. On the other hand, for regression, the model is focused on fitting the data points, so data points tend to lie more on bounded regions with finite function values. Additional results from these experiments are included in Appendix G.

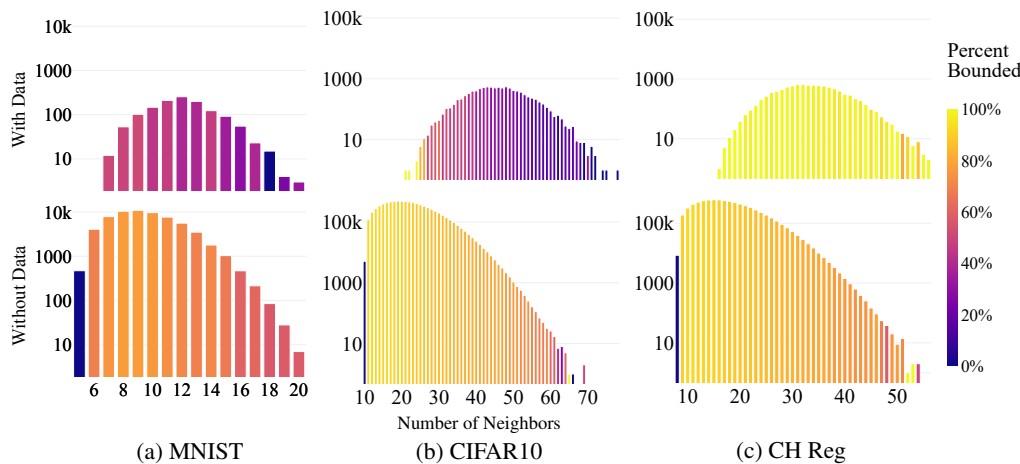

(a) MNIST  (b) CIFAR10  (c) CH Reg

Figure 7: Histograms of polyhedron neighbor counts, separated into those that contain data points (top) and those that do not (bottom). The vertical axis gives the number of polyhedra and each bar is colored by the percentage of polyhedra among those with the same neighbor count that are bounded.

## 6 DISCUSSION AND FUTURE WORK

This work characterizes general geometric properties of the polyhedral complexes defined by ReLU networks. For the first time, we place bounds on both the average connectivity of this complex and its graph-theoretic diameter. We also conduct empirical studies that visualize the distributions of polyhedron connectivity, and show that training data tends to lie on polyhedra with higher-than-average connectivity.

There are several limitations to the work presented here. Further investigation is needed to fully explain why training tends to put data points in regions with higher numbers of faces, and how this phenomenon is related to the network's behavior. Additionally, we are not yet able to describe how more specific network structures like convolutional layers and skip connections affect the network's geometry. Another limitation comes from the fact that our results only apply to ReLU activations, and while they could be extended to other piecewise-linear activation functions, there are no immediate implications for networks that use nonlinear activation functions.

Our results have implications for several active areas of study that involve the polyhedral geometry of ReLU networks. For example, the work by Ji et al. (2022) places a bound on empirical training error based on the spatial relationships between the regions containing the train and test data. They use Hamming distance between the sign sequences of two polyhedra as a distance metric between them. However, this metric will not reflect the case where a bent hyperplane may have to be crossed multiple times when moving from one polyhedron to another. Thus, the length of the shortest path between two polyhedra in the connectivity graph is a more suitable metric. If path length is used, Theorem 3.8 allows us to bound the empirical error based on the network architecture and independently of the input dimension.

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

APPENDIX

# A  DETAILED RELATED WORK

Several works have established lower and upper bounds on the *maximum* possible number of polyhedra (the regions in Fig. 1(b)) in ReLU-type networks (networks with piecewise-linear activation functions) in terms of input dimension, depth and width of the network (Pascanu et al., 2014; Montufar et al., 2014; Montúfar et al., 2022; Serra et al., 2018; Montúfar et al., 2022; Goujon et al., 2024). For example, when the hidden layer width is always larger than the input dimension, the number of linear regions grows exponentially with respect to both the input dimension and the number of hidden layers (Montufar et al., 2014). Another direction bounds from above the *expected* number of polyhedra over all networks of the same architecture (Hanin & Rolnick, 2019b). These bounds have been compared in a recent survey (Huchette et al., 2023), which summarizes how lower bounds are found by constructing networks with many polyhedral regions while upper bounds are found by determining the effect of adding additional layers of various sizes to existing networks. The number of polyhedra affects how the network transforms a trajectory in the input space after each layer of mapping (Hanin & Rolnick, 2019a), and the length of the transformed trajectory reflects the network expressivity (Raghu et al., 2017).

Several theoretical studies characterize the polyhedral structure of ReLU-like networks. A number of them have established links to tropical geometry (Zhang et al., 2018; Fournier, 2019; Charisopoulos & Maragos, 2019; Trimmel et al., 2021; Lezeau et al., 2024), which can be used to bound the total number of polyhedra and provide implicit formulas for the networks' decision boundaries. Analysis of ReLU networks as max-affine spline operators (Balestriero & Baraniuk, 2018; Balestriero & baraniuk, 2018; Balestriero et al., 2019) has identified a correspondence between the boundaries of the network's polyhedral regions as roots of a polynomial. Although this does not provide a direct way to see how the polyhedra fit together, this theory has been successfully applied in detecting toxicity in large language models by analyzing them individually (Balestriero et al., 2024). Topology has been essential for describing how the architecture of a network affects its geometry and measuring its complexity in terms of Betti numbers (Grigsby & Lindsey, 2022; Grigsby et al., 2024). An activation state is a sign vector indicating whether the ReLU in the corresponding hidden neuron has been activated for an input. A topological approach proves that there is a 1-1 correspondence between the network's activation states and its polyhedral regions and can calculate the polyhedron boundaries using this description of its combinatorial structure (Masden, 2025). This paper builds on top of this work, which constructs the robust theoretical framework we use to study ReLU geometry (see Section 2 for more details). A more efficient method for computing the boundaries is developed later on (Berzins, 2023), which builds on a previous method that can determine the architecture and parameters of ReLU networks via sampling (Rolnick & Kording, 2020). These works all provide tools for deeper analysis of network geometry, but they have not examined the connectivity of the polyhedra, a fundamental property of the complex.

There are also empirical studies that calculate the polyhedral complex and demonstrate a number of practical applications. Due to the intractability of enumerating all polyhedra, existing methods iteratively solve linear programs (LPs) that locally search confined regions of the input space (Liu et al., 2023a; Xu et al., 2022; Liu et al., 2023b) to count polyhedra or perform reachability analysis (Tran et al., 2019; Yang et al., 2020). The behavior of ReLU networks on bounded input spaces can also be exactly described with mixed-integer LPs with decision variables representing activation states, leading to a significant body of work improving the formulations of the networks with constraints (Huchette & Vielma, 2023; Tsay et al., 2021; Anderson et al., 2019) and applying them to problems such as verification (Botoeva et al., 2020; Cheng et al., 2017; Bunel et al., 2018) and inverse design (Ansari et al., 2022). Network architecture has been shown to influence the volume and distribution of the polyhedra (Zhang & Wu, 2019), which are then applied to quantify how the network generalizes to new data (Ji et al., 2022; Lehmann & Ebner, 2022; Daróczy, 2020). ReLU networks can be modified at the polyhedral level with the aim of improving explainability (Zhu et al., 2023; Hu, 2021). Other works study their geometric structure through the lens of activation states of the network on various inputs. For instance, a few methods use the Hamming distance between activation states as a proxy for similarity between the data points in different polyhedra, with various downstream applications ranging from data compression (He et al., 2021) to open set detection (Jamil et al., 2023) and clustering (Craighero et al., 2020). However, empirical studies

have not examined the connectivity of the polyhedral complex and how it relates to the distribution of training data.

## B  PROOFS OF THEORETICAL RESULTS

**Definition B.1** (Hypercube Graph). A $k$-hypercube graph is a graph formed from the vertices and edges of a $k$-hypercube. It can also be constructed as a graph with nodes corresponding to every string of $k$ bits and edges between nodes with strings that differ by one bit.

**Definition B.2** (Polyhedron). A polyhedron is a (possibly unbounded) intersection of halfspaces, or equivalently, the set of solutions to a system of linear inequalities of the form $\{x \in \mathbb{R}^d \colon \Phi x + \beta \leq 0\}$ for some real $(n, d)$-matrix $\Phi$ and some $\beta \in \mathbb{R}^n$.

**Definition B.3** (Polyhedral Complex). A Polyhedral Complex $\mathcal{C}$ is a set of polyhedra satisfying the following properties:

1. Closure under intersection, that is, $c_1, c_2 \in \mathcal{C}$ implies $c_1 \cap c_2 \in \mathcal{C}$. Note that this can be the empty set.

2. Closure under taking faces. That is, if $c \in \mathcal{C}$ and $c'$ is a face of $c$, then $c' \in \mathcal{C}$

We restrict our discussion to ReLU networks with the following two properties:

**Genericity** (Grigsby & Lindsey (2022), Definition 2.4): A hyperplane arrangement in $\mathbb{R}^d$ is called *generic* if all sets of $k$ hyperplanes intersect in an affine space of dimension $d - k$ for $1 \leq k \leq d$. In a ReLU network, each activation region is the intersection of half-spaces created by each neuron in the network, the boundaries of which form a hyperplane arrangement in the layer's input space. A neural network is generic if the hyperplane arrangements corresponding to each of its regions are generic.

**Supertransversality** (Masden (2025), Definition 11): Let $f$ be a ReLU network with complex $\mathcal{C}$ and denote by $f_i \colon \mathbb{R}^{m_{i-1}} \to \mathbb{R}^{\text{out}}$ the output of the last $\ell - i$ layers of $f$ with complex $\mathcal{C}_i \subset \mathbb{R}^{m_{i-1}}$. Denote by $H_i$ the polyhedral complex in $\mathbb{R}^{m_{i-1}}$ created by the hyperplane arrangement $\{x \in \mathbb{R}^{m_{i-1}} \colon \pi_j(W^{i-1}x + b) = 0\}$ where $\pi_j$ is the projection onto the $j$-th coordinate for some $1 \leq j \leq m_i$. Suppose for each layer $1 \leq i \leq \ell$ and each $1 \leq k \leq d$ that the restriction of $f_i$ to the interior of every $k$-cell of the complex of $H_{i-1}$ is transverse to the interior of all cells of $\mathcal{C}_i$. Then, the network $f$ is called *supertransversal*.

Define $N_k(\mathcal{C})$ to be the number of $k$-cells in the polyhedral complex $\mathcal{C}$.

**Theorem 3.4.** *The average number of faces of a $d$-cell of $\mathcal{C}$ in $\mathbb{R}^d$ (i.e., the average degree of the connectivity graph) is at most $2d$.*

*Proof.* A sign sequence that has exactly one $0$ in its elements corresponds to a $(d-1)$-cell. Each $(d-1)$-cell forms a boundary of exactly two $d$-cells in the polyhedral complex $\mathcal{C}$, and the single $0$ in its corresponding sign sequence can be set to $-1$ or $1$ to obtain the sign sequences of the two $d$-cells it separates. Thus, the sum of the numbers of faces of every $d$-cell is just twice the number of $(d-1)$-cells, and the average number of faces of a $d$-cell of $\mathcal{C}$ is $\frac{2N_{d-1}(\mathcal{C})}{N_d(\mathcal{C})}$. We want to show that $\frac{2N_{d-1}(\mathcal{C})}{N_d(\mathcal{C})} \leq 2d$, or equivalently, $N_{d-1}(\mathcal{C}) \leq dN_d(\mathcal{C})$. We perform mathematical induction on both the dimension of the (sub)complex $d$ and the number of neurons in the network $n$.

**Base Case**

If $d = 1$, each $d$-cell is a 1-cell, which can have either 0, 1, or 2 faces. Thus, the average number of faces of $d$-cells is at most 2.

If $n = 1$ (the network only consists of a single neuron), the arrangement consists of a single hyperplane, so the average number of faces of a $d$-cell is 1 and the lemma holds regardless of the value of $d$.

**Inductive Step**

We now prove that the statement holds for complexes of ReLU networks with $n$ neurons in $d$ dimensions if it holds for both $n-1$ neurons in $d$ dimensions and $n-1$ neurons in $d-1$ dimensions.

Assume for any ReLU complex $\mathcal{C}'$ with $n-1$ neurons (or $n-1$ BHs) in $d$ dimensions, we have

$$N_{d-1}(\mathcal{C}') \leq dN_d(\mathcal{C}'). \tag{4}$$

and for any ReLU complex $\mathcal{C}'$ with $n-1$ neurons in $d-1$ dimensions, we have

$$N_{d-2}(\mathcal{C}') \leq (d-1)N_{d-1}(\mathcal{C}'). \tag{5}$$

Let $\mathcal{C}$ be a ReLU complex of a network $f$ with $n$ neurons in $d$ dimensions. Let $h_i$ be the BH corresponding to a particular neuron $i$ in the last layer of $f$. By Lemma 3.3 where $k = d$,

$$N_d(\mathcal{C}) = N_d(\mathcal{C} - h_i) + N_{d-1}(h_i). \tag{6}$$

Because $\mathcal{C} - h_i$ is a complex with $n-1$ BHs in $d$ dimensions, Eq.(4) holds. Substituting Eq.(4) into Eq.(6) yields

$$N_d(\mathcal{C}) \geq \frac{1}{d}N_{d-1}(\mathcal{C} - h_i) + N_{d-1}(h_i). \tag{7}$$

Lemma 3.3 also applies to $N_{d-1}(\mathcal{C} - h_i)$ where $k = d-1$. Substituting it into Eq.(7) yields

$$N_d(\mathcal{C}) \geq \frac{1}{d}\left[N_{d-1}(\mathcal{C}) - N_{d-1}(h_i) - N_{d-2}(h_i)\right] + N_{d-1}(h_i),$$

which can be equivalently written as:

$$N_d(\mathcal{C}) \geq \frac{1}{d}N_{d-1}(\mathcal{C}) - \frac{1}{d}\left[N_{d-1}(h_i) + N_{d-2}(h_i)\right] + N_{d-1}(h_i). \tag{8}$$

To prove that $N_{d-1}(\mathcal{C}) \leq dN_d(\mathcal{C})$, it is equivalent to prove that $N_d(\mathcal{C}) \geq \frac{1}{d}N_{d-1}(\mathcal{C})$, and so it is sufficient to show that the remaining terms on the right-hand side of Eq.(8) are at least $0$. That is,

$$N_{d-1}(h_i) \geq \frac{1}{d}\left[N_{d-1}(h_i) + N_{d-2}(h_i)\right].$$

Multiplying this inequality by $d$ on both sides yields

$$dN_{d-1}(h_i) \geq N_{d-1}(h_i) + N_{d-2}(h_i).$$

Then, subtracting $N_{d-1}(h_i)$ from both sides yields

$$(d-1)N_{d-1}(h_i) \geq N_{d-2}(h_i).$$

Since $h_i$ is a ReLU complex in $d-1$ dimensions with $n-1$ neurons (its sign sequence contains only one $0$), the strict form of this inequality is guaranteed by Eq.(5). $\qquad\square$

**Theorem 3.1.** *For a ReLU complex in $d$-dimensional input space, the average number of faces of a $k$-cell is at most $2k$ for $k = 1, 2, \ldots, d$.*

*Proof.* For $k = 1, \ldots, d-1$, each $k$-cell of a polyhedral complex $\mathcal{C}$ corresponds to a sign sequence that has exactly $d-k$ zeros, so the points in the $k$-cell are contained in the intersection of these $d-k$ BHs. Each $k$-cell belongs to a unique subcomplex of sign sequences where the elements corresponding to these BHs (neurons) are fixed to $0$. Therefore, calculating the average number of faces of a $k$-cell involves examining the average number of faces of the $k$-cells contained in the subcomplexes restricted to every combination of $d-k$ neurons (i.e., every intersection of $d-k$ BHs and its corresponding subset of sign sequences with $0$ in the corresponding positions). These intersections are $k$-dimensional ReLU complexes themselves. Thus, Theorem 3.4 guarantees that the average number of faces for the $k$-cells in each intersection is at most $2k$. Together, the average number of faces of the $k$-cells contained in all different intersections of $d-k$ BHs is at most $2k$. $\quad\square$

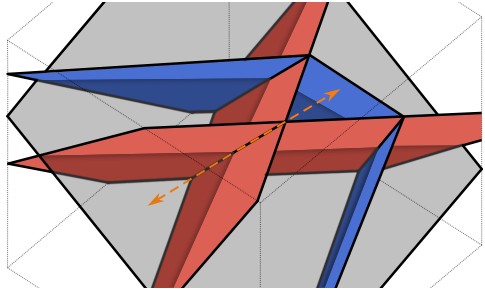

Figure 8: ReLU complex illustrating Theorem 3.5 with $d = 3$ and $n_1 = 2$, when there are no vertices in the complex. The first layer neuron's hyperplanes are shown in red, and the space orthogonal to their intersection ($\mathcal{R}(W_1)$) is shown in gray. A second layer BH is also shown in blue. Regardless of how we translate the gray plane, its intersection with all the BHs will be identical, and the blue second layer BH will never intersect the intersection of the two first-layer hyperplanes. The proof works by showing that as you move in a direction orthogonal to the intersection of the first-layer hyperplanes (shown in orange), the distance to the first layer hyperplanes stays the same, and so the output of the first layer (and thus, the remaining layers) stays the same.

Previous work gives a lower bound on the connectivity of the complex as follows:

**Lemma B.1** (Grigsby et al. (2024), Version 1, Corollary 5.29). *If a ReLU network has at least $d$ neurons in the first hidden layer, then every cell of $\mathcal{C}$ contains a vertex.*

If the condition of having at least $d$ neurons in the first layer is not met, the minimum degree can be lower. We generalize the previous result as follows:

**Theorem 3.5.** *If a ReLU network has $n_1$ neurons in the first hidden layer, every $d$-cell of $\mathcal{C}$ has at least $\min(n_1, d)$ neighbors, and thus the average degree of the connectivity graph is at least $\min(n_1, d)$.*

*Proof.* If $n_1 \geq d$, then Lemma B.1 implies every cell of $\mathcal{C}$ contains a vertex. Equivalently, every node in the connectivity graph is part of a $d$-hypercube graph, and the graph has minimum degree $d$. On the other hand, assume that $n_1 \leq d$. By the assumption of genericity, $W_1 \in \mathbb{R}^{n_1} \times \mathbb{R}^d$ has rank $n_1$. Thus, the space spanned by its row vectors $\mathcal{R}(W_1)$ (the normal vectors of the first layer's hyperplanes) has dimension $n_1$ and its null space $\mathcal{N}(W_1)$ has dimension $d - n_1$. Since the null space and row space of $W_1$ are orthogonal complements, every point $x \in \mathbb{R}^d$ can be written uniquely as $x = v + k$ for $v \in \mathcal{R}(W_1)$ and $k \in \mathcal{N}(W_1)$. However, since $W_1 k = 0$ for all $k \in \mathcal{N}(W_1)$, we really have that $W_1 x + b = W_1(v + k) + b = W_1 v + W_1 k + b = W_1 v + b$. This shows that for a fixed $v$, $f(x) = f(v + k)$ remains constant for all $k \in \mathcal{N}(W_1)$. It then immediately follows that no polyhedron boundaries are ever crossed when $k$ varies. Thus, to count the number of faces of $d$-cells in $\mathcal{C}$, we can count the number of faces of the corresponding $n_1$-cells of the complex $\mathcal{C}_{n_1}$ created by restricting the network to $\mathcal{R}(W_1)$. Applying Lemma B.1, every $n_1$-cell in $\mathcal{C}_{n_1}$ contains a vertex (0-cell). Every 0-cell in $\mathcal{C}_{n_1}$ corresponds to a $(d - n_1)$-cell in $\mathcal{C}$, which means every $d$-cell in $\mathcal{C}$ contains a $(d - n_1)$-cell. A $(d - n_1)$-cell's node in the connectivity graph belongs to a $n_1$ hypercube subgraph, which implies that the minimum degree of the connectivity graph is at least $n_1$ when $n_1 \leq d$. □

**Theorem 3.6.** *The average number of faces of $d$-cells in $\mathcal{C}_n$ increases monotonically in terms of $n$.*

*Proof.* To prove this, it is equivalent to prove that adding a new BH $h_n$ to the complex (i.e., adding a new neuron to the network) creates more $(d - 1)$-cells than $d$-cells, $N_{d-1}(\mathcal{C}) - N_{d-1}(\mathcal{C} - h_n) > N_d(\mathcal{C}) - N_d(\mathcal{C} - h_n)$. Lemma 3.3 with $k = d - 1$ shows that the left-hand side $N_{d-1}(\mathcal{C}) - N_{d-1}(\mathcal{C} - h_n) = N_{d-1}(h_n) + N_{d-2}(h_n)$. Applying the lemma with $k = d$ shows that the right-hand side $N_d(\mathcal{C}) - N_d(\mathcal{C} - h_n) = N_d(h_n) + N_{d-1}(h_n)$, but $N_d(h_n) = 0$, so the right-hand side of the equation just equals $N_{d-1}(h_n)$. Substituting these equations into the inequality on both sides yields $N_{d-1}(h_n) + N_{d-2}(h_n) > N_{d-1}(h_n)$. Then, subtracting $N_{d-1}(h_n)$ shows the inequality is equivalent to $N_{d-2}(h_n) > 0$. Since $(d-2)$-cells are created whenever $h_n$ intersects other BHs in the complex, which must happen at least once if any $d$-cells are created, so this inequality is true. Since

after the $d$-th term the sequence is monotonically increasing and bounded below and above by $d$ and $2d$ respectively, the average number of faces must converge to a value between these bounds. $\square$

**Theorem 3.7.** *Let $f$ be a shallow network that has only one hidden layer with $n$ nodes. When $n$ goes to infinity, the average number of faces of its $d$-cells converges exactly to $2d$. That is,*

$$\lim_{n \to \infty} \frac{2N_{d-1}(\mathcal{C}_n)}{N_d(\mathcal{C}_n)} = 2d.$$

*Proof.* If the network is shallow, the BHs of the neurons form a generic hyperplane arrangement. According to Buck (1943), the number of $k$-cells in the generic hyperplane arrangement $\mathcal{C}$ in $d$ dimensions is given by $N_k(\mathcal{C}) = \sum_{i=d-k}^{d} \binom{n}{i}\binom{i}{d-k}$. Intuitively, the first term in the sum counts subsets of hyperplanes that intersect in at least $k$ dimensions, and the second term counts the number of subsets in each of these sets intersect to form a $k$-face. Therefore, we have that

$$\frac{2N_{d-1}(\mathcal{C}_n)}{N_d(\mathcal{C}_n)} = 2\frac{\sum_{i=1}^{d} \binom{n}{i}\binom{i}{1}}{\sum_{i=0}^{d} \binom{n}{i}}$$

As $n \to \infty$, the numerator is dominated by $d\binom{n}{d}$ and the denominator is dominated by $\binom{n}{d}$. Therefore, the expression converges to $2d$. $\square$

Let $\ell$ be the total number of hidden layers (depth), $m_j$ be the number of nodes in layer $j$, $j = 1, \ldots, \ell$, and $m = \max\{m_1, \cdots, m_\ell\}$ (width).

**Theorem 3.8.** *The diameter $D$ of a ReLU complex's connectivity graph is $\Omega\left(\frac{\ln(N_d(\mathcal{C}))}{\ln(n)}\right)$ and $O\left(m^\ell\right)$.*

*Proof.* For the lower bound, we can use the fact that the maximum degree of the connectivity graph is the number of hidden neurons $n$, as a single BH cannot form more than one face of a polyhedron. The Moore bound (Diestel, 2017) implies that the number of vertices in the connectivity graph is at most $1+n+n(n-1)+\cdots+n(n-1)^{D-1} < n(n-1)^D$. We can rearrange this inequality as follows: $N_d(\mathcal{C}) < n(n-1)^D$ becomes $\ln(N_d(\mathcal{C})) < \ln(n(n-1)^D)$, and $n, D \geq 1$, so $D > \frac{\ln(N_d(\mathcal{C}))}{\ln(n(n-1))}$.

To get the upper bound, consider two $d$-cells $p_1, p_2 \in \mathcal{C}$. If the network only consists of one layer, we can easily find a path of polyhedra from $p_1$ to $p_2$ in this hyperplane arrangement by drawing a straight line from a point within $p_1$ to a point within $p_2$ and flipping the sign of each element in $\mathcal{S}(p_1)$ every time we cross its BH. Each flip then yields the sign sequence of the next polyhedron on the path, and since the BHs are just hyperplanes, the line is guaranteed to cross each of the BHs on the path and flip the corresponding sign only once. For deeper networks, the problem is more difficult because BHs are not necessarily hyperplanes, and we may have to cross the same one more than once to get from one polyhedron to another. For a neural network with 2 layers, consider the BHs of the second layer as subdivisions of the polyhedra of the hyperplane arrangement created by the first layer. While walking along a path between two of these layer-1 polyhedra, we have to cross some of these subdivisions. Now, if we want to find a path between two polyhedra $p_1$ and $p_2$ in this network, we can still find one that only crosses each first-layer BH at most once, because we can ignore the second layer and find a path between the polyhedra from the first layer's hyperplane arrangement that contains $p_1$ and $p_2$. Inside each of the $m_1 + 1$ polyhedra created from only the first layer's BHs, the segments of each BH from the second layer are contained by a hyperplane. Recursively treating them as their own hyperplane arrangement, we know we can get from one side of a first-layer polyhedron to the other by crossing each second-layer hyperplane at most once. Thus, the total number of first and second layer BHs we must pass through to get from $p_1$ to $p_2$ is bounded above by $(m_1 + 1) \times m_2$. For deeper networks, we can continue this recursion for each layer. Then, the maximum number of faces we would have to pass through to get from one to any other is upper bounded by $\prod_{1 \leq j \leq \ell} (m_j + 1) \leq (m + 1)^\ell$. The dominant term of $(m+1)^\ell$ as $m, \ell \to \infty$ is $m^\ell$, which shows that the connectivity graph diameter is $O\left(m^\ell\right)$. $\square$

This diameter upper bound is the tightest asymptotic bound possible without factoring in the dimension of the complex. For $x \in \mathbb{R}$, the following equations give a non-degenerate 1-dimensional network with width $m = 2$, depth $\ell$, and more than $m^\ell$ linear regions. Denote by $f_{i,j}$ the $j$th neuron of layer $i$ with $1 \le i \le \ell$ and $j \in \{1, 2\}$.

$$f_{1,1}(x) = \max(x + 1, 0)$$
$$f_{1,2}(x) = \max(2x - 1, 0)$$
$$f_{i,j}(x) = 1.1jf_{i-1,1}(x) - 1.1jf_{i-1,2}(x) - \frac{3j - 2}{4^{i-1}}$$
$$f_{\ell,1}(x) = f_r(f_{\ell,1}(x) - f_{\ell,2}(x))$$

See an interactive demonstration of this network here:

`https://www.desmos.com/calculator/rc6bkezojk.`

## C  ADDITIONAL CATEGORIZATION EXAMPLES

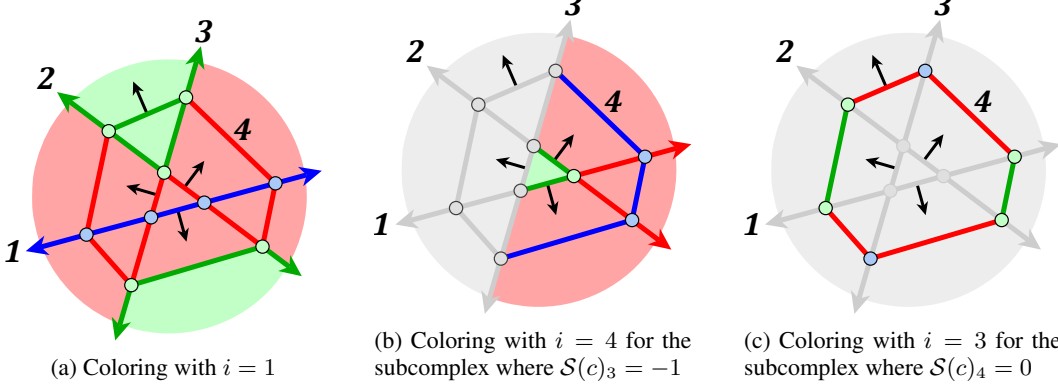

(a) Coloring with $i = 1$

(b) Coloring with $i = 4$ for the subcomplex where $\mathcal{S}(c)_3 = -1$

(c) Coloring with $i = 3$ for the subcomplex where $\mathcal{S}(c)_4 = 0$

Figure 9: Alternate categorizations for Fig. 3 with different choices of $i$ and restrictions to different subcomplexes of $\mathcal{C}$. Cells in Categories 1, 2, and 3 are shown in blue, green, and red, respectively. Cells that are not included in the complex are shown in gray. In Fig. 9c, the only $h_i$ cells are the blue vertices, and each one splits a pair of 1-cells in the subcomplex.

## D  ALGORITHMIC DETAILS

In this section, we describe and further explain the role of the subroutine SOLVELP() in Algorithm 1, which was executed by Gurobi[1] during our experiments. The SOLVELP() function takes in inputs $-\Phi_{s_i}$, $\Phi_s$, and $\beta_s + e_i$, which are calculated based on $s$ according to Eq. 2 and Eq. 3 to determine the affine functions forming the boundary of the corresponding polyhedron. To check if a linear inequality is redundant we solve the following LP for the best $x^*$ and we evaluate whether $\Phi_{s_i}^T x^* + \beta_{s_i} \le 0$.

---

[1] `https://www.gurobi.com`

$$
\begin{aligned}
\text{SOLVELP}(-\Phi_{s_i}, \Phi_s, \beta_s + e_i) = \text{maximize} \quad & -\Phi_i^T x \\
\text{subject to} \quad & \Phi_1^T x \leq -\beta_1 \\
& \Phi_2^T x \leq -\beta_2 \\
& \quad\quad \vdots \\
& \Phi_{i-1}^T x \leq -\beta_{i-1} \\
\text{Relaxed Constraint} \rightarrow \quad & \Phi_i^T x \leq -\beta_i + 1 \\
& \Phi_{i+1}^T x \leq -\beta_{i+1} \\
& \quad\quad \vdots \\
& \Phi_{n-1}^T x \leq -\beta_{n-1} \\
& \Phi_n^T x \leq -\beta_n
\end{aligned}
$$

Let $p \in \mathcal{C}$ be a $d$-cell with sign sequence $s$. Our goal is to find the neighbors of $p$, which correspond to $d$-cells with sign sequences that differ from $s$ in one position. If $p$ is separated by a face from the neighbor differing in sign sequence position $i$, this face will be contained by the BH of the neuron $i$ in the network. Therefore, to find the neighbors of $p$, we can equivalently determine which neurons have BHs that contain the faces of $p$, which we can do by checking each neuron individually. As described in Section 2, when $s$ is fixed and the network defines an affine function, the neuron $i$ defines a half-space (a linear inequality of the form $\Phi_{s_i}^T x + \beta_{s_i} \leq 0$) where the sign of its affine function is indicated by $s_i$. This half-space contains $p$ by definition, and the intersection of the half-spaces of all of the neurons in $f$ is exactly $p$. However, not all of the half-spaces contain faces of $p$, as some may be redundant. To test whether a specific neuron $i$'s BH forms a face of $p$, we push the boundary hyperplane of $i$'s half-space backwards along its normal vector and check whether that allows us to walk further in that direction before we leave one of the half-spaces. That is, following (Fukuda, 2004), we relax the less-than inequality corresponding to neuron $i$ by adding a small value to the right-hand side and find the point $x$ in the new system of inequalities that maximizes $-\Phi_{s_i}^T x$, where $-\Phi_{s_i}^T$ is the opposite direction of the normal vector of the hyperplane bounding the half-space. If $x^*$ violates the original inequality $\Phi_{s_i}^T x + \beta_{s_i} \leq 0$, then it is outside of $p$, so that half-space is not redundant and the face of $p$ contained by its boundary is a $(d-1)$-cell of $h_i$, which we can cross (equivalently, flip $s_i$'s sign) to get to a new polyhedron. If $x^*$ does not violate the original constraint, then the half-space defined by neuron $i$ is redundant and its BH does not contain a face of $p$, so flipping the sign of $s_i$ does not yield a neighbor of $s$ and may not even correspond to any element of $\mathcal{C}$.

## E  ADDING BENT HYPERPLANES TO THE DUAL GRAPH

Here we describe how the process of adding a new neuron to the network, going from $\mathcal{C}_n$ to $\mathcal{C}_{n+1}$ via the addition of BH $h_{n+1}$, can be represented in the connectivity graph. Each step is visualized in Fig. 10. When a new neuron's BH is added to an existing complex, it splits each of a connected set of existing $d$-cells in $\mathcal{C}_n$ (the red nodes identified in step 1) in half. We first disconnect the nodes representing $d$-cells getting split from the rest of the graph, which form a vertex cut (step 2). Then, we duplicate the induced subgraph of the vertex cut, with each copy representing the Category 3 $d$-cells in $\mathcal{C}_{n+1}$ formed on either side of the new neuron's BH (step 3). After that, we add a new edge between each corresponding node in the two copies, representing the Category 1 $(d-1)$-cell between each of those $d$-cells (step 4). Together, this set of new edges represents the added neuron's BH. We now have to add back the edges we deleted between the vertex cut and the rest of the graph, which represent Category 2 $(d-1)$-cells that separated the $d$-cells that got split from the $d$-cells that did not (step 5). For each edge we removed in the second step, we add a new edge from its original outside-the-cut node to one of the two copies of its original in-the-cut node, the copy representing the polyhedron in $\mathcal{C}_{n+1}$ that is on the same side of the new BH as the original outside-the-cut node. This demonstrates the monotonicity of the average number of faces as network size increases (Theorem

3.6), since every added node uniquely corresponds to at least one new added edge regardless of the value of $d$.

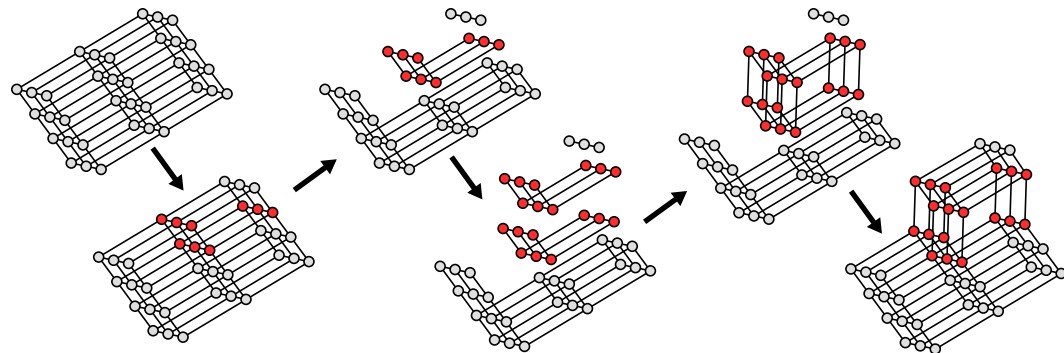

Figure 10: Connectivity graph modification when adding another final-layer neuron to a neural network with $d = 3$. Red nodes represent the $d$-cells that are split in half by a new neuron's BH. Note that the complex before and after is made up of cube graphs, and the subcomplex of split 3-cells is made up of square graphs (2-hypercubes).

# F  EXPERIMENTAL DETAILS

For all experiments, datasets were randomly split into 80% train and 20% test. Models were trained using vanilla SGD with a fixed learning rate. Data from all real datasets was normalized, and labels were normalized for the regression task. All networks were trained on a single GPU. Calculating the polyhedra in their complexes was distributed across 32 processors, and took up to twelve hours for the largest networks.

For the experiments on synthetic data, all networks were trained for 10 epochs with a batch size of 64 and a learning rate of 0.01. The datasets used in these experiments were generated using Scikit-Learn[2]. For the experiments with real data, setup information and performance metrics are listed in Table 2.

Table 2: Architecture, training hyperparameters, and performance of networks trained on real-world data. For the regression task, we report coefficient of determination $R^2$ and Mean Squared Error (MSE), while for the classification tasks, we report accuracy and Receiver Operating Characteristic Area Under the Curve (AUC) on the test set.

| Dataset Name | CA Housing | CIFAR10 | MNIST |
|---|---|---|---|
| Dataset Size | 20640 | 60000 | 70000 |
| Task | Regression | Classification | Classification |
| # Classes | NA | 10.00 | 10.00 |
| Batch Size | 64 | 4 | 64 |
| Epochs | 60 | 30 | 50 |
| Learning Rate | 0.00 | 0.01 | 0.10 |
| Input Dimension | 8 | 10 | 5 |
| Hidden Layer Sizes | (128) | (64, 64) | (8, 8, 8) |
| Test Accuracy | NA | 0.64 | 0.90 |
| Test AUC | NA | 0.94 | 0.99 |
| Test $R^2$ | 0.65 | NA | NA |
| Test MSE | 0.34 | NA | NA |

For MNIST and CIFAR10, we split the trained networks into two parts and regard the first part as a feature extractor. Our experiments are then performed on the second part, a classifier, which has a smaller input dimension. The California Housing dataset only has 8 features, so we regard the full network as the classifier. We report the input dimension and hidden layer sizes of the classifier

---

[2]https://scikit-learn.org

networks in Table 2. Working with the lower input dimension allows us to calculate a larger portion of the polyhedral complex associated with the network, which gives us a more complete picture of its structure. For the MNIST network, the feature extractor consists of a single fully connected layer with a ReLU activation. For the CIFAR10 network, the feature extractor consists of two convolutional layers with $5 \times 5$ kernels computing 6 and 16 features, both followed by a $2 \times 2$ max-pooling layer and a ReLU layer, before a final fully-connected layer with a ReLU activation.

## G  EXTENDED EXPERIMENTAL RESULTS

We performed additional experiments and included the related results here. Fig. 11 illustrates the same results in Fig. 5 but with the theoretically computed lower diameter bound on the x-axis instead of the upper bound.

Fig. 12 expands our results in Fig. 4 to include results for networks with input dimension equal to 2 and 3 and display the average number of neighbors for each combination of width, depth, and dimension averaged over the 5 trials.

Table 3 provides more detailed results for the number of polyhedra, the average degree, the average volume of each bounded polyhedron, the percentage of bounded regions among all regions, and the graph diameter for different networks with input dimension between 2 and 5, numbers of hidden layers between 1 and 4, and width in $\{4, 8, 16\}$. We do observe that the number of polyhedra exponentially increases with the number of hidden layers (depth) and with the number of neurons in each layer (width). However, the average degree quickly approaches the upper bound for each dimension.

Fig. 13 shows how the volume and inradius of bounded polyhedra in the MNIST complex are related to their numbers of neighbors and whether or not they contain data. Although inradius has been previously used to estimate the volumes of polyhedra in higher-dimensional complexes where exact calculations are intractable, this shows an example of a case where the two values are not closely correlated. Note that unbounded polyhedra are not included in this figure.

Fig. 14 shows how the distribution of neighbor counts in Fig. 6a changes over the course of training. After each epoch, we recalculate the entire complex and check the positions of the same 10,000 data points. In this instance, we observe that as the network trains, the data is gradually surrounded by higher numbers of polyhedra with relatively many neighbors.

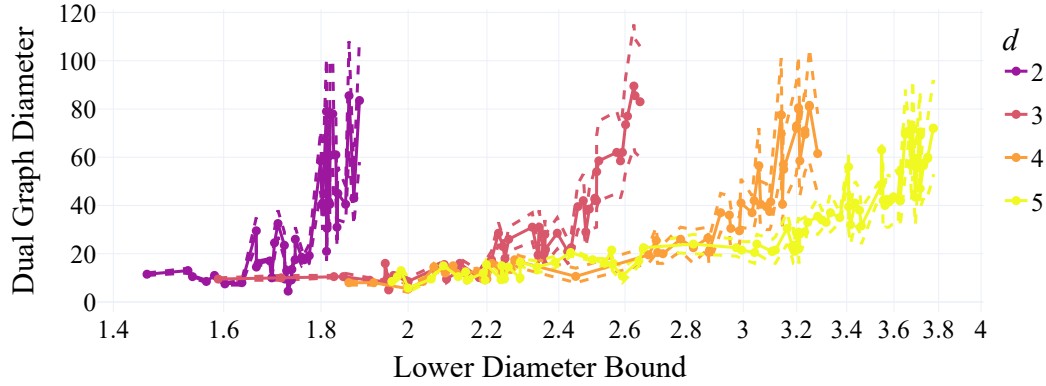

Figure 11: The same plot as Fig. 5 but with the lower bound from Theorem 3.8 on the $x$-axis and experiments from all widths plotted together.

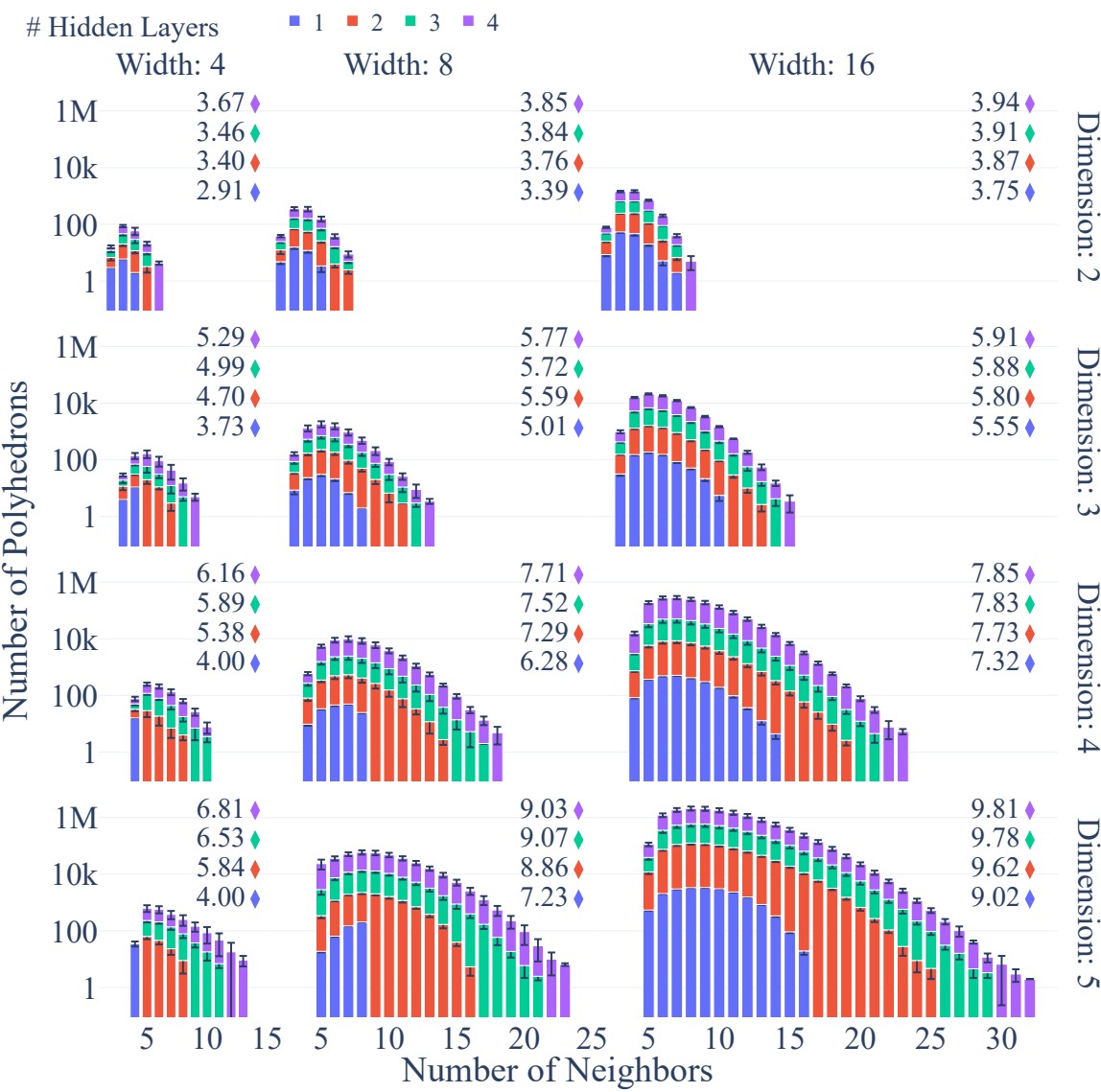

Figure 12: Distributions for the number of faces of polyhedra in the decompositions of trained ReLU networks with varying input dimension (row), width (column), and depth (color). Each colored bar shows the average number of polyhedra with a given number of faces created by a certain architecture across 5 runs, with error bars representing the standard deviation. Diamonds represent the average number of faces in each network.

Table 3: Summary statistics for the distributions in Fig. 12. Diameter for each complex is estimated as described in Section 5.1.

| $d$ | # Hidden | Width | # Regions | Avg # Facets | Average Volume | % Finite | Diameter |
|---|---|---|---|---|---|---|---|
| 2 | 1 | 4 | 11.00±0.00 | 2.91±0.00 | 0.63±0.11 | 0.27±0.00 | 4.50±0.00 |
| | | 8 | 37.00±0.00 | 3.46±0.00 | 1.45±0.99 | 0.57±0.00 | 9.10±0.22 |
| | | 16 | 137.00±0.00 | 3.74±0.00 | 3.34±4.48 | 0.77±0.00 | 19.20±0.76 |
| | 2 | 4 | 30.80±4.15 | 3.37±0.09 | 33.75±67.35 | 0.57±0.05 | 8.70±1.04 |
| | | 8 | 135.20±9.88 | 3.75±0.03 | 13.19±11.50 | 0.78±0.03 | 18.50±1.46 |
| | | 16 | 541.20±26.81 | 3.87±0.01 | 4.81±3.16 | 0.88±0.01 | 41.80±1.82 |
| | 3 | 4 | 58.20±14.58 | 3.52±0.10 | 116.74±147.94 | 0.67±0.07 | 11.70±1.44 |
| | | 8 | 275.20±51.57 | 3.82±0.02 | 45.21±33.08 | 0.84±0.02 | 27.10±3.45 |
| | | 16 | 1141.20±56.99 | 3.91±0.01 | 11.43±3.63 | 0.92±0.01 | 60.40±1.08 |
| | 4 | 4 | 87.80±22.86 | 3.63±0.09 | 153.52±121.53 | 0.81±0.03 | 14.40±2.22 |
| | | 8 | 497.20±148.28 | 3.85±0.03 | 109.09±28.45 | 0.86±0.03 | 37.00±5.72 |
| | | 16 | 2128.40±287.64 | 3.94±0.01 | 28.05±28.37 | 0.94±0.01 | 80.90±3.38 |
| 3 | 1 | 4 | 15.00±0.00 | 3.73±0.00 | 0.49±0.65 | 0.07±0.00 | 5.00±0.00 |
| | | 8 | 93.00±0.00 | 4.99±0.00 | 3.17±4.20 | 0.38±0.00 | 10.10±0.42 |
| | | 16 | 696.80±0.45 | 5.55±0.00 | 3.08±2.35 | 0.65±0.00 | 21.80±0.27 |
| | 2 | 4 | 62.00±9.77 | 4.68±0.17 | 5.07±5.18 | 0.45±0.05 | 8.80±0.45 |
| | | 8 | 710.00±141.76 | 5.56±0.06 | 20.26±16.75 | 0.68±0.03 | 20.20±1.68 |
| | | 16 | 5628.40±482.33 | 5.80±0.01 | 13.79±4.92 | 0.82±0.01 | 41.80±1.35 |
| | 3 | 4 | 118.80±49.34 | 4.91±0.25 | 60.28±67.19 | 0.60±0.07 | 11.00±1.17 |
| | | 8 | 1806.80±572.05 | 5.70±0.08 | 56.18±21.47 | 0.76±0.07 | 26.90±2.46 |
| | | 16 | $2.00\times10^4$±2871.17 | 5.88±0.01 | 30.68±12.59 | 0.89±0.01 | 59.00±3.30 |
| | 4 | 4 | 291.40±145.44 | 5.13±0.33 | 314.42±189.10 | 0.66±0.18 | 14.90±3.11 |
| | | 8 | 4073.40±1620.28 | 5.76±0.06 | 204.91±95.20 | 0.82±0.06 | 34.20±5.75 |
| | | 16 | $5.49\times10^4$±4259.65 | 5.91±0.02 | 91.56±49.71 | 0.91±0.02 | 81.70±6.45 |
| 4 | 1 | 4 | 16.00±0.00 | 4.00±0.00 | 0.00±0.00 | 0.00±0.00 | 5.50±0.00 |
| | | 8 | 163.00±0.00 | 6.28±0.00 | 4.78±9.59 | 0.21±0.00 | 10.60±0.42 |
| | | 16 | 2517.00±0.00 | 7.32±0.00 | 4.33±2.61 | 0.54±0.00 | 22.50±0.35 |
| | 2 | 4 | 72.60±22.70 | 5.21±0.31 | 16.85±37.67 | 0.33±0.18 | 8.70±0.76 |
| | | 8 | 2244.80±630.08 | 7.25±0.18 | 36.18±21.01 | 0.56±0.11 | 20.40±0.74 |
| | | 16 | $4.22\times10^4$±8608.12 | 7.72±0.06 | 13.07±4.95 | 0.78±0.05 | 41.10±0.65 |
| | 3 | 4 | 227.60±42.21 | 5.85±0.16 | 31.59±36.89 | 0.63±0.06 | 12.50±0.61 |
| | | 8 | 9340.80±3325.81 | 7.47±0.17 | 143.79±68.60 | 0.69±0.08 | 27.60±2.25 |
| | | 16 | $2.23\times10^5$ $7.21\times10^4$ | 7.82±0.04 | 49.91±9.37 | 0.85±0.04 | 57.70±2.46 |
| | 4 | 4 | 448.00±119.14 | 6.17±0.12 | 62.72±41.18 | 0.72±0.07 | 15.90±1.19 |
| | | 8 | $3.58\times10^4$±9493.85 | 7.70±0.06 | 233.75±72.31 | 0.85±0.02 | 37.40±1.29 |
| | | 16 | $6.24\times10^5$ $9.63\times10^4$ | 7.85±0.03 | 107.33±20.93 | 0.86±0.02 | 76.35±4.56 |
| 5 | 1 | 4 | 16.00±0.00 | 4.00±0.00 | 0.00±0.00 | 0.00±0.00 | 5.50±0.00 |
| | | 8 | 219.00±0.00 | 7.23±0.00 | 6.80±19.51 | 0.10±0.00 | 10.75±0.26 |
| | | 16 | 6884.87±0.35 | 9.02±0.00 | 4.35±2.69 | 0.44±0.00 | 23.17±0.41 |
| | 2 | 4 | 89.50±19.78 | 5.34±0.25 | 0.00±0.01 | 0.31±0.11 | 9.30±0.63 |
| | | 8 | 5802.60±1146.10 | 8.77±0.27 | 62.26±32.43 | 0.45±0.11 | 21.75±1.06 |
| | | 16 | $2.69\times10^5$ $4.87\times10^4$ | 9.61±0.05 | 21.69±10.54 | 0.71±0.04 | 42.57±1.53 |
| | 3 | 4 | 389.60±188.02 | 5.65±0.41 | 2.11±4.30 | 0.55±0.20 | 14.65±2.32 |
| | | 8 | $3.66\times10^4$ $1.21\times10^4$ | 8.54±0.93 | 156.80±75.65 | 0.68±0.12 | 31.50±2.33 |
| | | 16 | $1.78\times10^6$ $1.94\times10^5$ | 9.78±0.04 | 39.91±17.83 | 0.82±0.04 | 57.44±1.47 |
| | 4 | 4 | 1206.70±1154.00 | 5.50±0.78 | 1.38±1.78 | 0.77±0.16 | 18.60±3.98 |
| | | 8 | $1.82\times10^5$ $7.98\times10^4$ | 8.25±1.38 | 192.28±100.15 | 0.84±0.07 | 48.35±12.25 |
| | | 16 | $5.03\times10^6$ $1.07\times10^6$ | 9.80±0.03 | 152.54±58.24 | 0.84±0.03 | 70.88±1.19 |

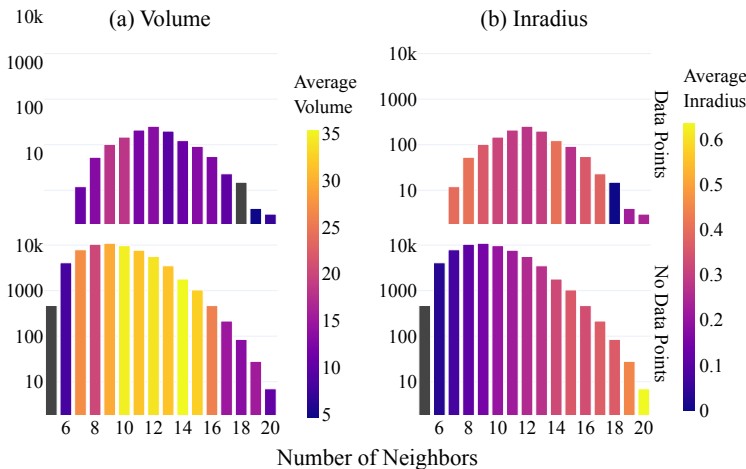

Figure 13: Histograms of *bounded* polyhedron neighbor counts for all $d$-cells in the MNIST network's complex, colored according to volume (left) and inradius (right).

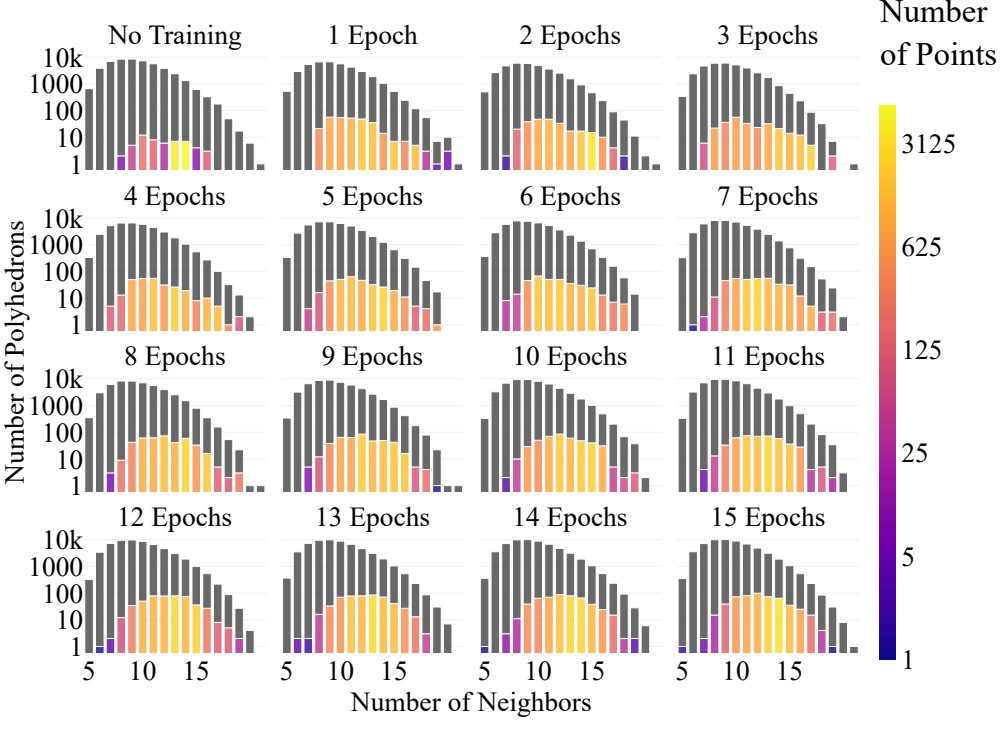

Figure 14: Face count distributions for a network trained on the MNIST dataset. These are constructed in the same way as in Fig. 6, but with one for each training Epoch.

