# OpenReview forum: "Characterizing the Discrete Geometry of ReLU Networks"
_ICLR.cc/2026/Conference — ICLR 2026 Oral_

### Official Review · Reviewer_drEY · 2025-10-19

**Soundness:** 3
**Presentation:** 4
**Contribution:** 3
**Rating:** 8
**Confidence:** 4

**Summary:**

This paper establishes fundamental results about the geometry of ReLU networks. It shows that the number of neighbors a linear region has is always upper bounded by 2 times the input dimension of the network. This is somewhat surprising because using a deep architecture can sometimes produce an exponential number of these regions, yet the configuration still obeys this graph property. The other main result is that the diameter of the connectivity graph has upper bounds that are independent of the input dimension, which is again surprising given the role the input dimension played in the first result.

**Strengths:**

1. Excellent presentation. This paper has great figures and clearly and concisely explains a lot of mathematical concepts.

2. Theoretical results were thought-provoking and intriguing, providing the reader with new ways to think about ReLU configurations.

3. Empirical results are well connected to the paper and make sense in light of the theory.

4. Theorem 3.6’s reasoning about connecting regions with lines does a good job explaining why the graph diameter really isn’t about the input dimension.

5. I was wondering about the connection with dividing space into cubes when I read the abstract and saw connectivity = 2d, it was cool to see that connection in Theorem 3.5

**Weaknesses:**

1. Line 80, missing the word ‘and’
2. 147, 169 ‘exactly’ should be ‘at least’ since a line can lie in the intersection of as many bent hyperplanes as it wants. It seems that most of the time we’re supposed to assume the BHs are in generic configuration though, mentioning that sooner could also fix it
3. 216 corresponding ‘to’ a network. ‘To’ is missing.
4. 269 ‘two d-ells’ typo
5. 316 ‘On a polyhedral region where the network’s behavior is affine.’
6. 408 complexes complexes
7. 825 appendix, is Lemma 3.4 supposed to be Theorem 3.4?
8. 839 ‘at once of the d-cells’
9. 863 did you mean to say each neuron adds d times more d-1 cells than d cells?
10. 936 contain should be contains

For Theorem 3.5, if the subgraph from the vertex cut has minimum degree d-1, doesn’t that mean that the number of new edges is only at least (d-1)/2 times the number of new vertices rather than (d-1)? I think this might affect the correctness of this proof. It would seem that if average degree (d-1) rather than 2(d-1) were all it took for a new (d-1) complex to contribute d times as many (d-1)-cells as d-cells to a d complex, that that would break the induction in theorem 3.4.

**Questions:**

One thing that confuses me about the proof of Theorem 3.4 (appendix A) is that it says it inducts over the ‘input dimension’ of the network, rather than the dimension of the ReLU complex. It establishes the base case as d=1, where all the 1-cells lie along a single line with two unbounded end segments, which is true of input dimension 1, but not generally true of an arbitrary 1-d ReLU complex, which could have multiple loops (such as the blue line in Figure 3b). It seems like it ought to be inducting over the ReLU complex dimension, since inductively adding neurons would need to check that the intersections of the existing BHs onto the new BH obey (d-2 cells) < (d-1)*(d-1 cells), which is different from having one less input dimension. Are you maybe using ‘input dimension’ to describe restriction onto a d-1 dimensional set of input space where the new neuron’s pre-activation output is 0? I guess I’m confused because the body of the paper makes the point that the dimension of the ReLU complex is more about the highest dimensional cells it has rather than the space it’s embedded in, but the induction seems to be over the embedding space. Some clarification here would be helpful.

---

> ### Author Response · Authors · 2025-11-22
>
> These comments were extremely insightful, we appreciate your help in improving this work.
>
> ## Weaknesses:
>
> We have made all of the minor changes you suggested in the updated manuscript.
>
> ### Theorem 3.5 (in the submitted version)
>
> Thank you for pointing out the error in Theorem 3.5, the fact that the edges were double counted does affect the correctness of the proof. Theorem 3.5 originally aims to show that the upper bound in Theorem 3.4 is tight. Therefore, we have taken the following steps to replace Theorem 3.5 and prove this tightness while preserving the majority of the original work:
>
> - Theorem 3.6 in the revised version contains our proof that the average degree of the connectivity graph monotonically increases in terms of the network size $n$, which was originally included as part of Theorem 3.5. This new version extends the monotonicity proof from our original submission from $d$-cells to all $k$-cells.
>
> -  We have identified a sequence of shallow ReLU network complexes where the average number of neighbors converges to exactly $2d$ when $n$ approaches infinity. Theorem 3.7 in the revised version uses this sequence to prove the tightness of our upper bound on the average number of neighbors of $d$-cells.
>
> - Lemma A.1 from the original submission has been replaced by Theorem 3.5 in the updated version since it complements our upper bound on the average number of neighbors by providing a lower bound of $d$ with the additional assumption that $n \geq d$.
>
> - To retain the original proof's link to cubes, we have moved our description of how adding BHs affects the connectivity graph along with the original figure to Section E of the appendix.
>
> ## Question about Theorem 3.4:
> Thank you for catching this discrepancy, we did mean induction on the dimension of the ReLU (sub)complex and not the input dimension of the network. 1-dimensional ReLU complexes can form loops like you described, and in that case every 1-cell has 2 faces, making the average degree exactly 2. This results in the change of all strict inequalities to inclusive inequalities in the proof.  We have made this change in the updated manuscript and also corrected the base case in the induction by simply stating that “1-cells have at most 2 faces."

---

> > ### Comment · Reviewer_drEY · 2025-11-23
> >
> > Thank you for your reply.  This paper was a pleasure to read.
> >
> > One thing that occurred to me after submitting the original review is that the upper bound on graph diameter is actually tight (line 321 says it’s loose). The description of a maximal diameter network from theorem 3.8 is realized by a one-dimensional relu network where each layer builds a triangle wave oscillating from 0 to 1 and back for each neuron. Each layer added to this network would insert m new points into each of the previous line segments, hitting the O(width^depth) bound.
> >
> > Also on line 914 latex rendered a ??

---

> > > ### Author Response · Authors · 2025-11-25
> > >
> > > We had not realized this, you’re absolutely right! We have constructed a network similar to the one you described and included a formula for it after our proof of the diameter bounds in the Appendix, along with an interactive visualization. We have also removed the error on line 914.

---

> > > > ### Comment · Reviewer_drEY · 2025-11-25
> > > >
> > > > Thank you; no further questions and I maintain my score/confidence (including in light of Fan et al., which I carefully read and assessed against the present manuscript; I feel similarly to reviewer MKyy).

---

### Official Review · Reviewer_5Du1 · 2025-10-30

**Soundness:** 4
**Presentation:** 4
**Contribution:** 2
**Rating:** 6
**Confidence:** 4

**Summary:**

The paper studies the geometry of piecewise-linear regions induced by ReLU networks, focusing on how these regions connect to each other rather than merely counting them. The authors define and analyze the connectivity graph, where nodes correspond to linear regions and edges represent shared facets between adjacent regions. They prove two main structural results. First, the average degree of this graph is at most $2d$, where $d$ is the input dimension, independent of network depth and width. Second, its diameter is upper-bounded by a quantity that depends only on the network’s width and depth, but not on input dimension. These bounds reveal a type of geometric regularity that holds across architectures.

On the algorithmic side, the paper proposes a method to enumerate regions and reconstruct the adjacency graph using linear programming feasibility checks. Experiments on small networks empirically validate the theoretical results and suggest that regions containing data points tend to have higher connectivity. Overall, the paper contributes new geometric understanding of ReLU networks, extending previous work.

**Strengths:**

The paper provides clear and rigorous geometric insights into the structure of ReLU networks by analyzing the connectivity of linear regions. The results are conceptually interesting, mathematically clean, and supported by both proofs and experiments. They are well presented and also lay a solid foundation for future work.

**Weaknesses:**

- The paper should explicitly state the basic assumption adapted from Masden (2022). Although it holds for a full-measure set, combinatorially it is arguably a strong restriction.
- While the paper gives helpful intuition and pictorial explanations, including a rigorous definition of the canonical polyhedral complex (at least in the appendix) would improve clarity.
- The results are not particularly deep; nonetheless, they are conceptually interesting, clean, and provide meaningful insights into neural network structure.

Minor:
- In the proof of Theorem 3.1, Lemma 3.4 is referenced but does not appear in the paper.

**Questions:**

- Is the connectivity graph the 1-skeleton of the sign sequence complex, which under the weight assumptions forms a cubical complex? Would your results hold more generally for graphs that are the 1-skeleton of a cubical complex? Have you checked the literature for related results on 1-skeleta of cubical complexes?
- How exactly do you define a ``face''? It seems that you mean a codimension-1 face, which is usually called a facet.
- Can you say anything about the non-generic case? Understanding the class of possible connectivity graphs for networks with 2 hidden layers would be very interesting.
- The observation that data points lie in cells with more neighbors is interesting. Do you have any idea why this might be the case? It has also been observed that data points tend not to lie close to the boundaries of cells after successful training (e.g., https://openreview.net/forum?id=NpufNsg1FP). Do you see any connection between these observations?

---

> ### Author Response · Authors · 2025-11-22
>
> Thank you for taking the time to review our work, below are our responses to your questions and concerns.
>
> ## Weaknesses
>
> - **State Assumptions:**  Detailed statements of the assumptions used in Masden, 2022 are now provided in Appendix A.
>
> - **Define Polyhedral Complexes:** A formal definition of a polyhedral complex has also been added to Appendix A.
>
> - **Incorrect Lemma Reference:** This reference has been corrected to the proper theorem.
>
> ## Questions
>
> 1. Your characterization of the sign sequence complex is accurate. Not all of our results hold for general cubical complexes, for example in complexes corresponding to ladder graphs (e.g., the complex ⧻ with dual graph ▥), the average number of neighbors does not approach $2d$ as the length of the ladder increases towards infinity. However, a shallow ReLU network with such a connectivity graph would not obey our “generic” assumption since some of its BHs would be parallel hyperplanes. As of now, we are not aware of any literature on 1-skeletons of cubical complexes that helps to further characterize ReLU networks.
>
> 2. We do define “faces” as facets, as stated in the Polyhedral Complex subsection of the Preliminaries section. Outside of polyhedral geometry, the term “face” is more commonly used to describe what are technically the facets of 2D and 3D objects, rather than the objects themselves. We chose this terminology to leverage readers’ existing intuition within the machine learning community. However, we recognize that “face” and “facet” are not interchangeable in certain mathematical contexts. To address this, we have added a note in the Preliminaries section of the revised manuscript, placed next to our first definition of “face”.
>
> 3. Without the “generic” assumption, the connectivity graph is not necessarily cubical, as more than $d$ BHs can intersect at a point. In our paper, we cite [Balestriero et al., 2019](https://dl.acm.org/doi/10.5555/3454287.3455705), which shows that complexes of ReLU networks are power diagrams. Power diagrams are dual to weighted Delaunay triangulations, and there is some existing work characterizing these as graphs. For example, [Bose et al., 2011](https://doi.org/10.1016/j.jda.2011.03.001) show that they have spanning ratios bounded by constants.
>
> 4. Please see our overall response above. There might be a connection between our observation and the observation that data tend to lie far from cell boundaries because for polyhedra of a given volume, the minimum distance from the Chebyshev center to the boundary can increase with additional facets.

---

### Official Review · Reviewer_6xrb · 2025-10-31

**Soundness:** 3
**Presentation:** 4
**Contribution:** 3
**Rating:** 8
**Confidence:** 4

**Summary:**

This paper provides new results on the geometric characterisation of the polyhedral complex generated by the linear regions of ReLU neural networks.  The theoretical results provide, in some sense, a universality result in that the bound on the degree of the connectivity graph does not depend on depth and width of the network.  Experimental results are provided alongside.

**Strengths:**

The paper is very well written.  The presentation is very nice and the ideas and concepts are presented very well.

**Weaknesses:**

- Limitations of the work were not discussed.
- Although there is quite a number of references cited and a further detailed section on related work in the appendix, I still felt that there wasn't a strong connection between the contributions of the submission and previous work.  For example, the cited work by Zhang et al. on the tropical geometry of neural networks provides a theoretical upper bound on the number of linear regions.  I think that the submission could have done some computational comparisons on the proposed approach versus this existing well-known tropical geometry paper.

**Questions:**

What can be said about the computational complexity of the approach versus some recent work, e.g., by Stargalla et al. "The Computational Complexity of Counting Linear Regions in ReLU Neural Networks"?  Also to other existing work, e.g., by Lezeau et al. "Tropical Expressivity of Neural Networks" that gives the exact linear region count as well as discusses geometry of the linear regions?

---

> ### Author Response · Authors · 2025-11-21
>
> We appreciate your comments and suggestions, please see our responses below.
>
> ## Weaknesses
>
> - **Limitations:** We have added a discussion of the limitations of this work to the conclusion section.
>
> - **Comparisons with Existing Work:** Our literature review broadly covers articles related to the polyhedral structure of ReLU networks. Prior works established several upper bounds on the number of linear regions such as the one you mentioned, Zhang et al., 2018, but our paper is one of the first to examine connections between the regions instead of simply counting them. It is difficult to computationally compare our work to Zhang et al., 2018 since that paper does not describe a method for enumerating polyhedra or calculating their connectivity and requires that network weights must be scaled to integers. From a theoretical perspective, our papers derive bounds on different quantities (e.g., graph connectivity rather than number of polyhedra). We will also include a discussion linking our results to the relevant prior work cited by Area Chair 5kuQ (Fan et al, 2024).
>
> ## Questions
>
> - As stated in Stargalla et al., 2025, deciding whether a network has more than K regions is an NP-hard problem. The comparable algorithm for counting activation regions is actually a brute force search, used to prove that the problem is in PSPACE. The algorithm in our work is consistent with their findings.
>
> - Thank you for pointing out the work by Lezeau et al., 2024, have added it to our related work section. The paper does not formally discuss the computational complexity of their method, and the analysis is not trivial since it depends on the decomposition of the input network into a difference of tropical polynomials. However, in the code of their implementation, they calculate the intersection of two polyhedra for each activation region. This is [roughly equivalent](https://people.inf.ethz.ch/fukudak/polyfaq/node25.html) to our most expensive step of determining redundant inequalities from a system of linear inequalities for each activation region (cf. Sec. 4.1).
> Other existing algorithms may be even faster at enumerating polyhedra in a given network, such as the one we cited in [Berzins, 2023](https://proceedings.mlr.press/v202/berzins23a.html), and we plan to take advantage of these in future work.

---

### Official Review · Reviewer_MKyy · 2025-11-01

**Soundness:** 4
**Presentation:** 3
**Contribution:** 3
**Rating:** 8
**Confidence:** 3

**Summary:**

ReLU networks partition a $d$-dimensional input space into linear regions and therefore have an associated canonical *polyhedral complex* in which the $d$-cells correspond to individual linear regions and lower-dimensional cells arise from their iterative intersections.
By considering only top level polyhedra ($d$-cells) as nodes and their faces ($(d-1)$-cells) as edges, the polyhedral complex can be simplified into a *connectivity graph*, retaining information of neighoring linear regions.
The core contributions of the paper are to prove theoretical results: (1) an upper bound on the average number of neighbors of $k-cells$ in the complex, (2) a convergence result towards this upper bound for $d-cells$ with the increase of network size (number of neurons) and (3) lower and upper bounds on the diameter of the connectivity graph.
Empirical experiments confirm these results and, in addition, reveal an intriguing observation: linear regions that contain data points tend to be more connected i.e. have more neighboring regions, than those that do not.

**Strengths:**

The paper appears technically sound and is written in a clear, engaging style, with informative and visually appealing figures and illustrations (except Figure 6; see my specific comment below).
The overall structure and section breakdown are logical, and the paper’s main message is conveyed smoothly. Experimental details are provided in the appendix, and code availability supports reproducibility.

The work contributes new theoretical results in an active research area. It complements existing studies on counting linear regions by considering instead their adjacency relations.
This perspective has ties with other work branches such as tropical geometry of ReLU networks and error bounds.
While the immediate actionable implications of the theoretical results are not obvious, they could inform studies on expressivity or serve as guideline to use quantities like degree of the connectivity graph as features.
Each claim is well supported both by proofs and empirical validation and the paper never overreaches.

While not new the polyhedra search method is interesting on its own.

**Weaknesses:**

The paper feels somewhat incomplete on one point.
The observation that polyhedra containing data tend to have, on average, more neighbors than those that do not is intriguing and calls for further exploration.
It would be helpful to confirm this finding with additional experiments to confirm it is not an artifact.
Intuitively, one might expect that regions containing data are more constrained and thus require higher expressivity and more complex boundaries to better fit the data.
While a theoretical explanation may be out of reach, additional empirical checks would, in my opinion, strengthen the paper, for instance:
- Check that this distribution difference emerge gradually during training by measuring the same statistics as in Fig. 6 at multiple training steps (including initialization).
- If region sizes can be estimated, are data-containing regions systematically smaller (indicating higher flexibility) ?
- Equivalently, is the spatial density of regions correlated with data density ? For example by taking a representative point by linear regions and measuring overlap with train/test data.
- Could the observed difference partly reflect the presence of peripheral, unbounded regions that would tend to be empty?

## Minor and additional feedback
- the links don't work
- line 76: unclear definition of the diameter
- typo missing "to" line 216
- line 825 lemma 3.4 does not exist in the paper
- strange syntax line 316
- vizualizing the lower bound or the number of regions $N_d(\mathcal C)$ would be nice on fig 5
- Fig 6 is a bit confusing: if the gray bars represent polyhedra traversed by BFS and colored bars are the subset of these polyhedra containing at least 1 data point, why is are gray bars below the colored ones on the right tail of plots (b) and (c) ?

**Questions:**

1. **Neighbor Count Distribution and Training Data** One of the most intriguing findings is the shift in the distribution of neighbor counts between regions that contain training data and those that do not. Do the authors have an intuition (or theoretical explanation in simple cases) for why regions containing data tend to have higher connectivity? (see also my comment above)
1. **Expressivity and Local Degree** To what extent could the number of neighbors in the connectivity graph be interpreted as a local measure of expressivity? On one hand more neighboring regions suggest finer partition of the input space, but on the other hand a simplicial partition of the input space would have maximal connectivity graph degree $d+1$ while still being able to be a good approximation (e.g. 3D meshes).
1. **Bounded vs unbounded regions** There is no discussion of bounded vs unbounded (peripheral) linear regions and while it does not change theoretical results they coexist implicitly throughout the paper. Moreover the unbounded regions can have less neighoring regions as you note in the proof of Theorem 3.4 in appendix for the base case, so I was curious to know if you had some general comment on this distinction and whether it is a useful framing.
1. **Diameter Bound and Implicit Dependence on Input Dimension** One of the hypothesis of theorem 3.6 is "as many first-layer neurons as input dimensions" and the derived upper bound is in $O(m^l)$. In the limit $d\mapsto \infty$ this implies $m=m_1\geq d$, which seems to be inconsistent with the claim of line 309 (also in the abstract) that the upper bound does not depend at all on the input dimension d. Could you clarify this point ?

---

> ### Author Response · Authors · 2025-11-21
>
> Thank you very much for your feedback, we tried to address all of your comments here.
> ## Weaknesses and Questions
>
> We have been able to perform several of the experiments you described.
>
> - **Q1 and Q2 - Neighbor count distribution and expressivity:** Please see our overall response for a detailed discussion of our results.
>
> - **Q3 - Bounded vs. unbounded regions:** We ran experiments looking at bounded and unbounded regions for all our networks trained on real-world data and our results were mixed. For the two classification tasks, a high proportion of the data appeared on unbounded polyhedra, but on the regression task, the data appeared more on bounded polyhedra. We have added these figures to Appendix G.
>
> - **Q4 - Condition on the first hidden layer:** The condition of “as many first-layer neurons as input dimensions" is not needed for the proof of Theorem 3.6 (now Theorem 3.7), so this diameter bound actually does not rely on input dimension. In our updated version, we have removed this sentence.
>
> - **Are data-containing regions smaller?**  Given our observations so far, this seems to not be the case.  Based on the experiments with bounded/unbounded regions, data can stay in many unbounded regions for classification tasks, so data-containing regions are not necessarily small.  Calculating the exact volume of the bounded regions becomes intractable very quickly (we calculated volumes of regions up to 5 dimensions). Region volumes in ReLU Networks have previously been estimated using inradius [(Zhang & Wu, 2020)](https://arxiv.org/abs/2001.01072). However, we performed additional experiments to compare exact volumes to inradii in 5 dimensions (see Figure 12 in Appendix G) and found that in our case they did not correlate.
>
> ## Minor and Additional Feedback
>
> All minor comments have been addressed.
>
> - Figure 5 did not have much room, so instead of adding the lower bound to it we opted to add Figure 9 to Appendix G that plots the same data as Figure 5 except with the lower bound on the x-axis instead of the upper bound.
>
> - Figure 6 and its caption have been updated for clarity. In the original plot, if a polyhedron was found by BFS but had data points in it, it was not included in the gray bars. When most of the polyhedra with a certain number of faces contain data points, the gray bar will be smaller than the colored bar in the corresponding position. In the updated manuscript, we stacked the bars in the chart and updated our caption to make this figure more clear. We were also able to improve our code’s efficiency after our initial submission and expand our BFS search from 1,000,000 polyhedra to 8,000,000 polyhedra.

---

> > ### Comment · Reviewer_MKyy · 2025-11-25
> >
> > Thank you for your answers and the additional experiments. At the moment I have no further questions.
> >
> > On the comparative analysis with Fan et al, I think that while the motivation is indeed close but the results and extensions proposed in the current paper add significant value and the authors convincingly addressed the potential overlap; therefore I maintain my score and good opinion of the paper.

---

### Comment · Area_Chair_5kuQ · 2025-11-19
**Request for Comparative Analysis with arXiv:2305.09145**

Dear Reviewers,

Thank you for your diligent work on reviewing this submission. As we proceed with the discussion, I would like to draw your attention to a potential overlap between the main contribution of this paper—specifically, the results presented in Theorem 3.1—and those established in an existing arXiv preprint:
https://arxiv.org/pdf/2305.09145 (see in particular Theorems 8–12).

While the two works may differ in motivation, technical framing, or application context, the core conclusions appear notably similar. I encourage all reviewers to carefully compare the two sets of results and assess the degree of novelty and incremental contribution of this submission.

In addition, I request that the authors provide a clear explanation of how their work differs from the aforementioned preprint, and to what extent their results build upon or diverge from prior findings. A detailed comparison in the rebuttal would be greatly appreciated.

Thank you for your attention to this matter.

AC

---

> ### Author Response · Authors · 2025-11-20
>
> Thank you for bringing this paper to our attention. A first look at the work by Fan et al., 2024 shows that it does also examine the connectivity of polyhedra by bounding the average number of faces. However, further examination reveals that our paper differs significantly from this prior work both overall and specifically in the area of potential overlap between their Theorems 8-12 and our Theorems 3.1 and 3.4, which we will focus on here.
>
> ## Assumptions
>
> Theorems 8 and 12 in Fan et al., 2024 assume a single hidden layer. In this case, the network’s complex is simply a hyperplane arrangement, and a stronger bound of exactly $2d$ was already proven in [Fukuda et al., 1991](https://doi.org/10.1016/0166-218X(91)90067-7). Theorem 9 handles networks of multiple layers but assumes that the input dimension $d=2$, which makes the analysis considerably easier. Theorems 10 and 11 assume that the network does not have bias terms, a strong assumption that forces all BHs to pass through the origin. Theorems 11 and 12 both assume that the first hidden layer’s weight matrix has rank $d_0$ which is less than the input dimension, and Theorems 8-12 all assume that the network’s input space is bounded within a hypercube $[-B,B]^d$. In contrast, our theorems concerning the average number of faces in the complex do not constrain the structure of the network in any of these ways.
>
> ## Asymptotics in Fan et al., 2024
>
> Apart from the simpler case of Theorem 9, the bounds in Fan et al., 2024 are all asymptotic (they include terms that are big O functions of $n$, the number of hidden units in the network). Thus, when the network size varies, the bound varies. Even removing this asymptotic term from their bounds by assuming the network has a certain number of neurons (leading to $2d+1$ in Theorem 8, $3d-1$ in Theorem 10, $2d_0+d-1$ in Theorem 11, and $2d_0 +1$ in Theorem 12), our bound of exactly $2d$ is tighter without limiting network structure. Our results in Theorems 3.1 and 3.4 provide a clear view that average connectivity is bounded independently of network size.
>
> ## Method of Analysis
>
> Crucially, the line of thinking in their proofs is completely different from ours. Their proofs never refer to bent hyperplanes, the actual geometric form of the activation state boundaries of neurons in later hidden layers that makes deep networks so expressive. In deep networks, only the first hidden layer neuron boundaries are hyperplanes, and the activation boundaries for neurons in subsequent layers "bend" whenever they intersect BHs from previous layers, which dramatically increases geometric complexity. Although multi-layer networks are still analyzed in Theorems 10-11, the assumptions that the network has zero biases and/or that the first hidden layer’s weights maintain a low rank $d_0<d$ simplify the geometry enough so that the proofs can directly apply a face-counting theorem for simple hyperplane arrangements. In contrast, our proofs work directly with bent hyperplanes, allowing for a unified argument for networks with arbitrary depths, widths, and biases.
>
> Our results are also more general. In Theorem 3.1 we extend to all $k$-cells of ReLU complexes instead of just the maximal cells (i.e., $d$-cells), and we then provide bounds on diameter and a lower bound on $d$-cell connectivity to complement the upper bound. Taken together, our results offer a more comprehensive characterization of ReLU complexes.
>
>
>
> ## Algorithms and Experiments
> Due to the computational complexity of enumerating activation regions and counting their faces, the work by Fan et al., 2024 simplifies the computation by employing Monte-Carlo methods in both finding polyhedra and counting their faces. Thus, their results are only approximations. In our experiments, we do not simplify the calculations in this way, and our algorithm systematically identifies all network regions and faces, which is extremely important if there is any correlation between region size and number of faces. For example, if more densely connected regions tended to be larger, their method of searching by calculating the polyhedra that contain uniformly sampled points would be biased towards higher face counts. Although they test networks with larger widths, Figures 4-7 show that they are only sampling a fraction of the polyhedra in the complex, whereas almost all of our experiments involve complete enumeration.
>
> Thus, our submission differs substantially from the work of Fan et al., 2024. Still, this work is clearly highly relevant, and we will add a thorough discussion to the end of our introduction section to properly acknowledge its contributions and distinguish them from our own work.

---

### Author Response · Authors · 2025-11-21

We appreciate the thoughtful feedback from our reviewers and area chair, it has been extremely helpful in improving our paper. Here we address common questions, and then we will provide responses to each individual reviewer to address their specific concerns. Unless explicitly stated otherwise, all changes that we discuss in our responses have already been made to the manuscript, which we have uploaded to OpenReview for you to view.

### Common question: Why does data tend to lie on polyhedra with higher-than-average connectivity, and how is this observation is related to expressivity?

Our intuition about this question aligns with that of Reviewer MKyy. Higher region density and connectivity both require a higher density of BHs and their intersections. Since the neural network must concentrate the BHs around the data during training, more highly connected regions may naturally arise around them.

As Reviewer MKyy suggested, we have trained a ReLU network on MNIST and used our Algorithm 1 to generate histograms of neighbor count distributions after each epoch. We observed in this experiment that data points shift to higher-connected polyhedra in general over the course of training. This result has been included in Figure 13 in Appendix G.  We will continue to run related experiments during the rebuttal period to gain additional insight into this observation.

There may also be other ways to explain the observation that region connectivity is higher around training data. For instance, there is also evidence from Dhayalkar, 2025 that the diameter of the connectivity graph is related to the VC dimension of the network, which suggests that controlling the connectivity of linear regions may control the expressivity of the model.

---

### Meta-Review · Area_Chair_vFiC · 2026-01-02

**Summary:**

The submission studies the connectivity graph defined by adjacency between linear regions in the input space induced by ReLU networks. It shows that the average degree is bounded by twice the input dimension, independent of network depth and width, and establishes diameter bounds whose upper bound is independent of the input dimension. These results are supported by experiments on synthetic and real data, together with an empirical observation that regions containing data tend to exhibit higher connectivity.

Reviewers commended the clarity and quality of the presentation, noting that the central geometric insights are articulated in a clean, conceptually appealing, and thought-provoking manner. With the initial concerns addressed, the paper stands as a solid and well-executed contribution to the geometric understanding of ReLU networks.

**Reviewer Concerns:**

The rebuttal addressed the main reviewer concerns, including novelty and overlap with prior work, clarification of assumptions and definitions, correction of technical issues, and additional empirical validation. No substantial concerns remain outstanding (beyond acknowledged limitations).

**Reviewer Scores:**

Reviewers have either reaffirmed their high scores or are likely to maintain their original evaluations.

---

### Decision · Program_Chairs · 2026-01-26

Accept (Oral)